# Long-Context Modeling with Dynamic Hierarchical Sparse Attention for On-Device LLMs

## Abstract

The quadratic cost of attention hinders the scalability of long-context LLMs, particularly in resource-constrained settings. While attention is known to be often sparse, existing static sparse methods such as sliding windows or global tokens cannot adapt to task or input dependent variations in attention. While there are recently proposed dynamic approaches for sparse attention, they still depend on predefined templates or heuristic mechanisms that reduces generality and may prune tokens that remain contextually important. As such, we introduce **Dynamic Hierarchical Sparse Attention (DHSA)**, a data-driven framework that dynamically predicts attention sparsity online without retraining of the base LLM. Our proposed DHSA adaptively segments sequences into variable-length chunks, then computes chunk representations by aggregating the token embeddings within each chunk. To avoid the bias introduced by varying chunk lengths, we apply a length-normalized aggregation that scales the averaged embeddings by the square root of the chunk size. Finally, DHSA upsamples the chunk-level similarity scores to the token level to produce importance scores that determine which token-level interactions are preserved. Our experiments with Needle-in-a-Haystack and LongBench show that DHSA preserves near-dense accuracy even in highly sparse regimes, yielding 12–20% relative accuracy gains over Block Sparse Attention at comparable prefill cost. Using a FlashAttention-2 (FA2)-based kernel, DHSA achieves up to a $10\times$ prefill speedup over dense FA2 at 128K context length. On Llama-3.1-8B (4-bit), DHSA scales to 100K context with strong accuracy and competitive latency on a single 24 GB GPU, where dense kernels fail between 16K and 64K. The implementation supports both GPU and CPU backends and is compatible with multiple open-weight model families. These results highlight DHSA as an efficient and adaptable solution for long-context inference for on-device LLMs.

## 1 Introduction

Long-context modeling expands the capabilities of large language models (LLMs) to a wide range of real-world applications (Yang et al., 2024), including proofreading and summarizing long documents, analyzing personal histories, and maintaining extended context in personal assistants. However, the quadratic complexity of the attention mechanism (Vaswani et al., 2017) leads to prohibitive computational costs, limiting the scalability of long-context tasks, especially for on-device inference.

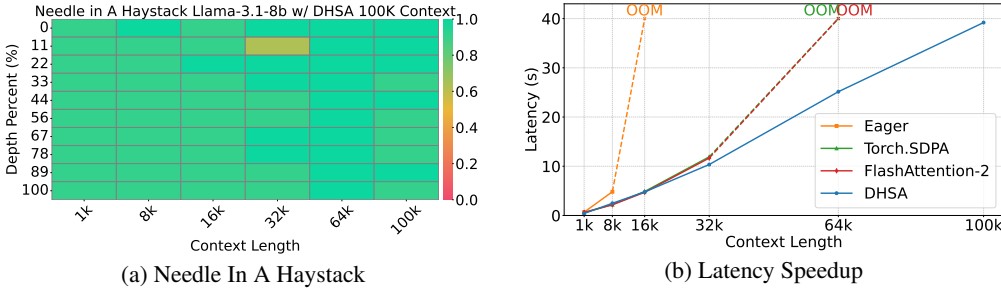

(a) Needle In A Haystack
(b) Latency Speedup

Figure 1: Needle-in-a-Haystack and end-to-end prefill latency on LLaMA-3.1-8B (4-bit) with DHSA. In (a), with a fixed density of 6.25%, it maintains high retrieval accuracy up to 100K context. In (b), DHSA achieves competitive prefill latency on a single NVIDIA RTX 3090 (24 GB), remaining feasible where dense kernels fail.

Table 1: Comparison of sparse attention methods. We evaluate whether each method accelerates *prefill*, supports *all model families*, and is *hardware-compatible (HW)*, i.e., runs on both GPU and CPU. [†]Conceptually architecture-agnostic, though current open-source implementations support only a subset of LLM families.

| Method | Prefill | All Models | HW |
|---|---|---|---|
| StreamingLLM | | ✓ | ✓ |
| MInference | ✓ | ✓ | |
| Block-Sparse | ✓ | | |
| DuoAttention | | ✓[†] | |
| SeerAttention | ✓ | | |
| Quest | | ✓[†] | |
| **DHSA (ours)** | ✓ | ✓ | ✓ |

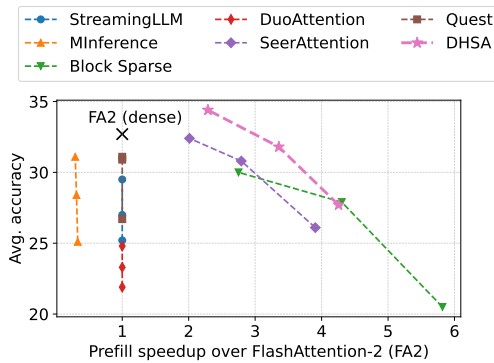

Figure 2: Accuray and (kernel-level) prefill speedup trade-off on LongBench with LLaMA-3.1-8B (4-bit).

Prior research has shown that attention matrices in LLMs are highly sparse (Child et al., 2019), motivating the development of static sparse attention methods such as Longformer (Beltagy et al., 2020) and BigBird (Zaheer et al., 2020) for more efficient inference. However, later studies demonstrated that attention distributions vary significantly across different tasks and inputs (Jiang et al., 2024). Such dynamic nature of the attention patterns limits the effectiveness of static sparse methods in long-context settings, as they often incur noticeable performance drops (Figure 13). Consequently, if the sparse attention patterns could be predicted in an input-adaptive dynamic manner, long-context LLMs can substantially reduce resource computational costs while maintaining accuracy.

Recent work has started to explore this idea through approaches that target either the prefill stage or the decoding stage. MInference (Jiang et al., 2024) accelerates prefilling by applying pre-defined sparsity patterns: it calibrates a template on sample data and then approximates sparsity indices using partial queries and keys. In contrast, H2O (Zhang et al., 2023) addresses decoding by introducing key–value cache eviction policies, estimating token importance from accumulated attention scores with hand-crafted rules. While shown to be effective in certain cases, both methods share a key limitation: they depend on manually designed patterns or heuristics, which restrict the search space of possible sparse structures and overlook the highly input-dependent nature of attention sparsity. This highlights the need for more flexible, data-driven approaches to sparsity prediction.

To address these challenges, we propose **Dynamic Hierarchical Sparse Attention (DHSA)**, an *inference-time, drop-in sparse attention module* for standard decoder-only Transformers. DHSA requires *no retraining of the base LLM*: given per-layer queries/keys, it predicts a token-level sparsity mask that the attention kernel uses to prune compute in both prefill and decode. Unlike template- or heuristic-based methods, DHSA learns content-adaptive sparsity online from token embeddings, delivering speedups across tasks without manual tuning.

Our contributions are summarized as follows:

- **Hierarchical routing formulation.** We recast sparsification as *chunk-level routing*: predicting a chunk–chunk similarity matrix and upsampling it to token-level importance in order to preserve only the most impactful query–key pairs while maintaining causal semantics (§3.1).

- **Content-aware segmentation with length-robust representations.** A lightweight boundary predictor uses token keys to *dynamically chunk* sequences. The predictor is shared across layers and adds only a linear-time pass in sequence length. Each chunk is then encoded with a length-normalized pooling scheme to produce stable chunk queries/keys for reliable similarity estimation (§3.2, §3.3).

- **Practical model- and hardware-agnostic sparse attention module.** We implement DHSA as a drop-in module that interposes between queries/keys and the attention kernel, performing efficient token-level selection. The implementation supports all model families and runs on GPUs and CPUs without retraining the base LLM (§3.4).

Extensive experiments on Needle-in-a-Haystack (Kamradt, 2023) and LongBench (Bai et al., 2023) show that DHSA preserves near-dense accuracy even in highly sparse regimes, yielding 12–20% relative accuracy gains over Block Sparse Attention (Han, 2024) at comparable prefill cost (Figure 2).

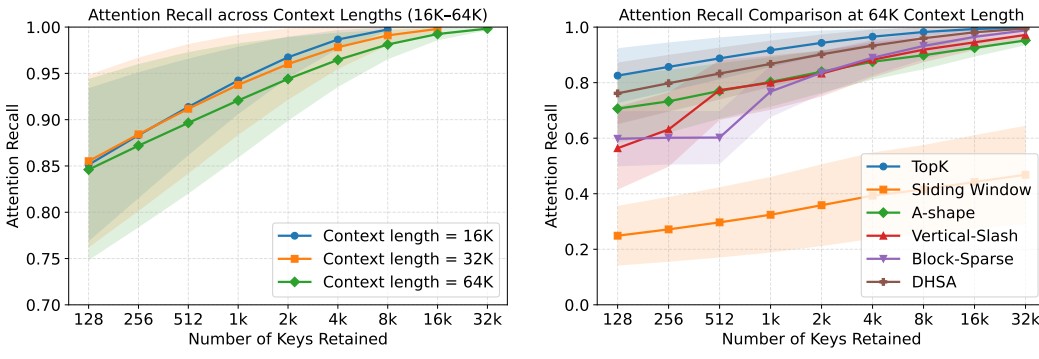

(a) Attention Recall with TopK Keys      (b) Attention Recall across Sparsity Patterns

Figure 3: We analyze attention sparsity on LLaMA-3.1-8B (4-bit). (a) More than 95% of the attention mass can be recovered by retaining only a small subset of keys. (b) DHSA consistently achieves higher recall than the other sparse patterns. See Appendix C and Figure 12 for theoretical analysis and additional empirical comparisons.

The implementation supports both GPU and CPU backends and is compatible with multiple open-weight model families (Table 1). These results highlight DHSA as an efficient and adaptable solution for long-context inference for on-device LLMs.

## 2 PREDICTING ATTENTION SPARSITY IN LONG CONTEXTS

**Long-Context Attention is Sparse** The quadratic cost of attention makes scaling LLMs to long contexts prohibitively expensive, especially in resource-constrained settings. However, a closer examination of attention distributions shows that much of this computation is unnecessary (Child et al., 2019). For example, as shown in Figure 3a, preserving only a small fraction of keys retains more than 95% of the total attention mass in LLaMA-3.1-8B (4-bit). In practice, each query token attends to only a limited subset of the sequence. These findings indicate that efficient long-context inference is achievable by accurately identifying and retaining the most important token interactions.

**Attention Sparsity Requires Input-Adaptive Prediction** Although long-context attention is sparse, the locations of salient tokens vary significantly across inputs, as relevance depends on the current query. Existing approaches often rely on static templates (e.g., sliding window (Beltagy et al., 2020), A-shape (Han et al., 2023a)) or predefined heuristics (e.g., Vertical-Slash (Jiang et al., 2024)), which limit performance on various tasks. At the same time, attention maps reveal structural regularities: salient tokens cluster into spans such as sentences, paragraphs, or code blocks. However, existing dynamic methods (e.g., Block-Sparse (Han, 2024)) disregard this structure by partitioning sequences into fixed-length chunks.

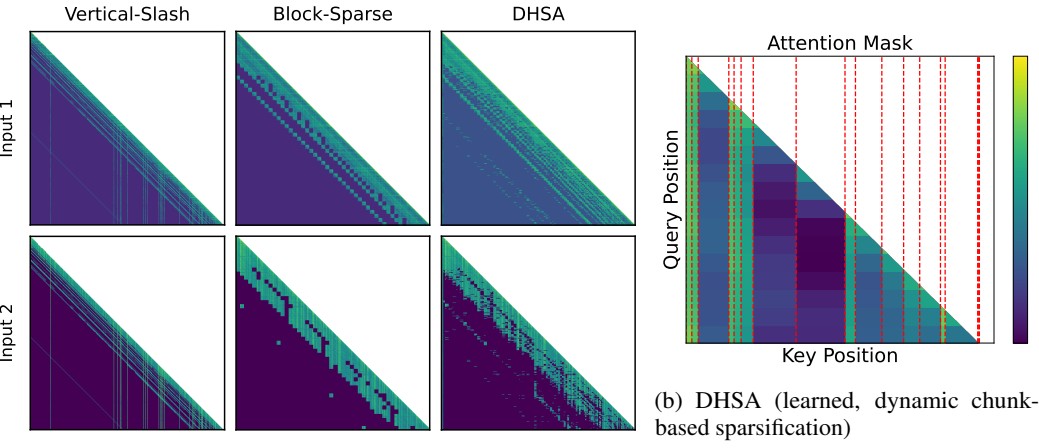

(a) Attention Sparsity Patterns      (b) DHSA (learned, dynamic chunk-based sparsification)

Figure 4: (a) Attention sparsity visualization for LLaMA-3.1-8B (4-bit). Existing methods such as Vertical-Slash and Block-Sparse enforces rigid regions. In contrast, DHSA produces adaptive masks across inputs. (b) DHSA introduces dynamic chunking (red dashed lines), offering greater flexibility and yielding better performance.

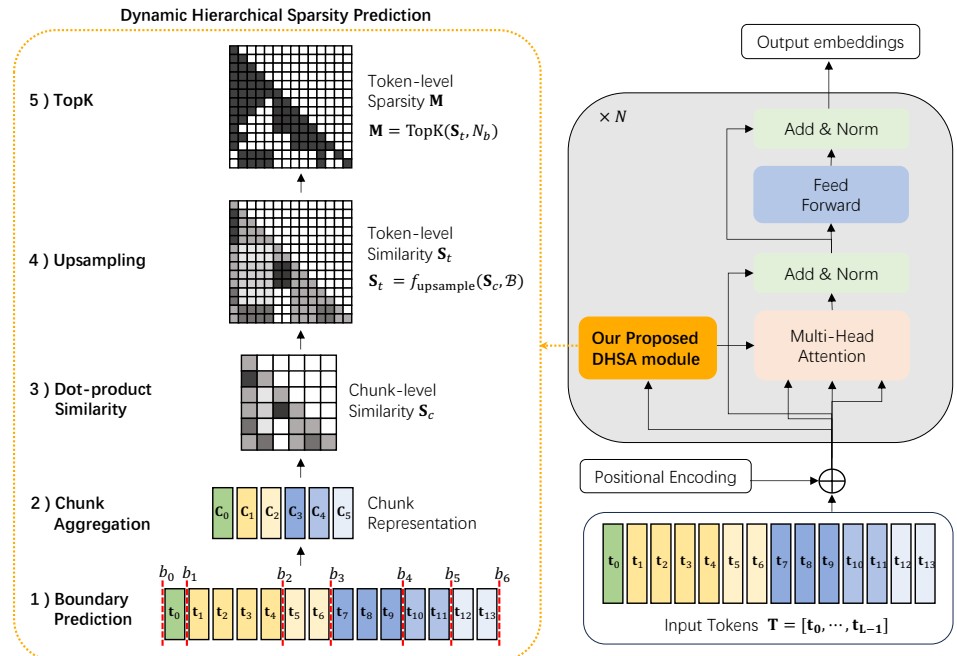

Figure 5: Overview of our proposed Dynamic Hierarchical Sparse Attention (DHSA) framework.

To this end, we propose **Dynamic Hierarchical Sparse Attention (DHSA)** that leverages the hierarchy that occurs from the structural regularities of attention maps for more efficient and performative long context modeling. DHSA predicts the importance of input tokens online by first performing dynamic chunking (example shown in Figure 4b) based on our proposed dynamic boundary detection method (Section 3.2) and then using chunk-level similarity to guide token-level sparsity (Section 3.3). As shown in Figure 4a, DHSA produces more flexible and adaptive patterns than existing methods. Furthermore, our comparisons in the attention recall shown in Figure 3b demonstrate that DHSA consistently achieves higher recall and closely approaches TopK performance.

## 3 DYNAMIC HIERARCHICAL SPARSE ATTENTION

Our proposed DHSA is a drop-in module integrated into each of the $N$ Transformer layers of an LLM (see Figure 5). It takes the token embeddings at the current layer as the input and outputs a sparsity mask to prune less important token pairs. To enable efficient prediction, we first group consecutive tokens into *chunks*. The core idea is to leverage **chunk-level similarity** to inform **token-level sparsity** prediction. This requires addressing two key challenges: (1) fixed-size chunking is too rigid to capture **content shifts** across varying inputs, and (2) average pooling poorly handles **variable-length chunks**. We resolve these with the solutions detailed in Sections 3.2 and 3.3, and validate that DHSA addresses both challenges in our ablation study (Table 4). First, in Section 3.1 we give an overview of the steps for hierarchical sparsity prediction with preliminaries and notations.

### 3.1 PRELIMINARIES

Given a sequence of tokens $\mathbf{T} = [\mathbf{t}_0, \mathbf{t}_1, ..., \mathbf{t}_{L-1}]$ of total length $L$, its token-level sparsity mask is denoted as $\mathbf{M} \in \{0, 1\}^{L \times L}$, where if the $i^{th}$ row and $j^{th}$ column element of $\mathbf{M}$ is equal to 1, i.e., $\mathbf{M}_{i,j} = 1$, it indicates that the interaction between $\mathbf{t}_i$ and $\mathbf{t}_j$, $i, j \in [0, L-1]$ should be preserved. Conversely, $\mathbf{M}_{i,j} = 0$ implies that the interaction between $\mathbf{t}_i$ and $\mathbf{t}_j$ can be skipped.

Predicting the full matrix $\mathbf{M}$ directly would require scoring all $L \times L$ token pairs, which is computationally prohibitive for long contexts. Instead, we adopt a two-step hierarchical approach:

**Step 1 Chunk-level prediction**: We partition the entire token sequence $\mathbf{T}$ into $N_c$ non-overlapping chunks $\{\mathbf{C}_0, \mathbf{C}_1, ..., \mathbf{C}_{N_c-1}\}$, defined by the boundary indices $\mathcal{B} = \{b_0, b_1, ..., b_{N_c}\}$, where $0 = b_0 < b_1 < ... < b_{N_c} = L$. Each chunk $\mathbf{C}_k$, $k \in [0, ..., N_c - 1]$ contains the consecutive tokens indexed from $b_k$ to $b_{k+1}$. We then construct a chunk-level similarity matrix

$\mathbf{S}_c \in \mathbb{R}^{N_c \times N_c}$, where the $l^{th}$ row and $k^{th}$ column element of $\mathbf{S}_c$, i.e., $(\mathbf{S}_c)_{l,k}$ represents the predicted importance of interactions between chunks $\mathbf{C}_l$ and $\mathbf{C}_k$. The procedure for how we divide the chunks and obtain $\mathbf{S}_c$ will be described in the subsequent Sections 3.2 and 3.3.

**Step 2 Token-level selection**: Starting from $\mathbf{S}_c$, we upsample it to obtain the token-level similarity matrix $\mathbf{S}_t \in \mathbb{R}^{L \times L}$, which encodes the predicted importance of attending from each query token to every key token. Concretely, for each chunk pair $(\mathbf{C}_l, \mathbf{C}_k)$, $\{l, k\} \in [0, N_c - 1]$, the corresponding submatrix $(\mathbf{S}_t)_{[b_l:b_{l+1}],[b_k:b_{k+1}]}$ is assigned the same value $(\mathbf{S}_c)_{l,k}$. We define such mapping function as $f_{\text{upsample}}(\mathbf{S}_c, \mathcal{B}) := \mathbf{S}_t$. Given the token-level similarity matrix $\mathbf{S}_t$, we generate the token-level sparsity mask $\mathbf{M}$ by applying a TOPK selection with a per-query token budget defined as $N_b$. Here $N_b$ is a hyperparameter, which may be set dynamically based on available computational or memory resources.

By effectively dividing the sequence of tokens in to chunks based on their similarity (described in Sections 3.2 and 3.3) and utilizing $\mathbf{S}_c$ to predict $\mathbf{M}$ through $\mathbf{S}_t$, we substantially decrease the cost for having to directly predict the full token-level sparsity matrix as shown in Figure 1b and Table 5. Next, in Section 3.2 we provide the details on how we determine the boundary indices $\mathcal{B}$ for the chunks.

## 3.2 DYNAMIC BOUNDARY DETECTION

To effectively estimate the token-level similarity matrix $\mathbf{S}_t$ from the chunk-level similarity matrix $\mathbf{S}_c$, we propose a dynamic chunking strategy that adaptively determines boundary indices $\mathcal{B}$ based on the input sequence $\mathbf{T}$. We formulate chunking as a **boundary detection** problem, where the goal is to decide whether each token position marks the *end of a chunk*. Formally, for each position $i \in [0, L-1]$, we define a boundary indicator function $\delta(i) = 1$ if $i = b_k$ for some $k$; otherwise, $\delta(i) = 0$. We estimate this indicator using a neural network with three components also shown in Figure 6:

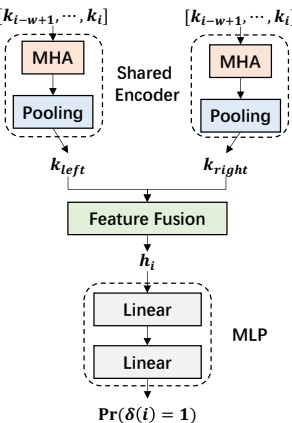

**Encoder.** For each candidate position $i$, we extract two local windows:

$$
\begin{aligned}
\mathbf{k}_{\text{left}} &= f_{\text{MHA}}([\mathbf{k}_{i-w+1}, \cdots, \mathbf{k}_i]), \\
\mathbf{k}_{\text{right}} &= f_{\text{MHA}}([\mathbf{k}_{i+1}, \cdots, \mathbf{k}_{i+w}])
\end{aligned}
\tag{1}
$$

where $w$ is the window size and $\mathbf{k}_i$ denotes the key vector of token $i$. Each window is processed by a Multi-Head Attention (MHA) module with pooling (denoted as $f_{\text{MHA}}$).

Figure 6: Architecture of the boundary predictor, consisting of a shared encoder, a feature fusion layer, and an MLP.

**Feature Fusion.** Given $\mathbf{k}_{\text{left}}$ and $\mathbf{k}_{\text{right}}$, we construct the feature vector:

$$
\mathbf{h}_i = [\mathbf{k}_{\text{left}}, \mathbf{k}_{\text{right}}, |\mathbf{k}_{\text{left}} - \mathbf{k}_{\text{right}}|, \mathbf{k}_{\text{left}} \odot \mathbf{k}_{\text{right}}, \text{sim}(\mathbf{k}_{\text{left}}, \mathbf{k}_{\text{right}})]
\tag{2}
$$

where $\odot$ is element-wise multiplication and $\text{sim}(\cdot, \cdot)$ is cosine similarity. Further rationale and analyses of this fusion choice are provided in Appendix B.2.

**MLP.** The fused feature $\mathbf{h}_i$ is passed through two linear layers defined as $f_{\text{MLP}}$, yielding the probability that position $i$ is a boundary:

$$
\Pr(\delta(i) = 1) = f_{\text{MLP}}(\mathbf{h}_i)
\tag{3}
$$

**Pipeline.** Given the sequence $\mathbf{T} = [\mathbf{t}_0, \ldots, \mathbf{t}_{L-1}]$, we extract their key vectors $\mathbf{K} = \{\mathbf{k}_i\}_{i=0}^{L-1}$. For each position $i$, the predictor takes as inputs two local context windows $\mathbf{K}_{i-w+1:i}$ and $\mathbf{K}_{i+1:i+w}$, and outputs a probability $\Pr(\delta(i) = 1)$. By applying non-maximum suppression step to these probabilities, we obtain the boundary set $\mathcal{B} = \{b_0, b_1, ..., b_{N_c}\}$, where each $b_k$ marks the end of a chunk. Due to space constraints, we provide the training and inference details for our boundary predictor in Appendices B.3 to B.5.

## 3.3 ROBUST CHUNK REPRESENTATION

Now that we have the boundaries $\mathcal{B}$, we aggregate the token embeddings to form chunk representations. We identify two main challenges arise in this process: (1) average pooling after padding is problematic,

as zero embeddings from padding dilute the average, and (2) average pooling is sensitive to chunk length. To address these issues, we compute the prefix sum of embeddings and divide by the actual chunk length, followed by **length normalization**:

$$\mathbf{q}_{\mathbf{C}_k} = \frac{\sqrt{b_{k+1} - b_k}}{b_{k+1} - b_k} \cdot \sum_{i \in [b_k, b_{k+1})} \mathbf{q}_i, \quad \mathbf{k}_{\mathbf{C}_k} = \frac{\sqrt{b_{k+1} - b_k}}{b_{k+1} - b_k} \cdot \sum_{i \in [b_k, b_{k+1})} \mathbf{k}_i, \qquad (4)$$

where chunk $\mathbf{C}_k$ spans the token interval $[b_k, b_{k+1})$, and $\mathbf{q}_i, \mathbf{k}_i$ are the token-level query and key vectors, respectively. Stacking the chunk representations yields $\mathbf{Q}_c = [\mathbf{q}_{\mathbf{C}_1}, \ldots, \mathbf{q}_{\mathbf{C}_{N_c}}]^\top$ and $\mathbf{K}_c = [\mathbf{k}_{\mathbf{C}_1}, \ldots, \mathbf{k}_{\mathbf{C}_{N_c}}]^\top$. The chunk-level similarity matrix is then given by $\mathbf{S}_c = \mathbf{Q}_c \mathbf{K}_c^\top$.

## 3.4 Accelerating LLM Inference through DHSA

DHSA is designed to primarily accelerate the prefill stage (see Algorithm 1), where the attention cost scales quadratically with context length. For completeness, we also specify a **decode-time variant** that bounds the keys attended by each newly generated token (see Algorithm 2). Cost analysis for DHSA is provided in Appendix C.6.

**Prefill stage.** When all tokens in the prompt sequence of length $L$ are available, we first predict the boundary indices $\mathcal{B}$ for the entire prompt. We then perform chunk-level prediction to obtain $\mathbf{S}_c$, upsample it to produce the token-level similarity matrix $\mathbf{S}_t$ with $f_{\text{upsample}}$, and apply TOPK selection with budget $N_b$ to obtain the token-level sparsity mask $\mathbf{M}$ for all tokens in the prompt.

**Decode stage.** In autoregressive generation, we adapt the approach to handle the incremental arrival of new tokens. Let $L'$ be the total sequence length at the current decoding step (including both prompt and generated tokens). We extend the existing chunk boundaries $\mathcal{B} = [b_0, ..., b_{N_c}]$, $b_{N_c} = L$ to $\mathcal{B}' = [b_0, ..., b_{N_c}, L' - 1, L']$. Here we have, total number of chunks $N_c + 2$, chunk $\mathbf{C}_{N_c} = [\mathbf{t}_L, ..., \mathbf{t}_{L'-2}]$ which contains all previously generated tokens in the current decoding session. and chunk $\mathbf{C}_{N_c+1} = [\mathbf{t}_{L'-1}]$ that contains only the current query token. We then compute only the interactions between $\mathbf{C}_{N_c+1}$ and all its preceding chunks to obtain last row of the updated chunk-level similarity matrix $\mathbf{s}_{c,\text{new}}$. This row is upsampled to the token level to produce the corresponding row of the updated token-level similarity matrix $\mathbf{s}_{t,\text{new}}$, from which we derive the last row of the sparsity mask $\mathbf{m}_{\text{new}}$ by applying TOPK selection with token budget $N_b$.

---

**Algorithm 1** DHSA: Prefill Stage

**Require:** prompt tokens $\{\mathbf{t}_i\}_{i=0}^{L-1}$ of length $L$; prompt chunk boundaries $\{b_j\}_{j=0}^{N_c}$ of length $N_c + 1$; per-query token budget $N_b$
**Ensure:** token-level sparsity mask $\mathbf{M}$
1: Define chunks $\mathbf{C}_j \leftarrow [\mathbf{t}_{b_j}, \ldots, \mathbf{t}_{b_{j+1}}]$, for $j = 0$ to $N_c - 1$
2: Obtain chunk-level queries $\mathbf{Q}_c[j, :]$ and keys $\mathbf{K}_c[j, :]$ for each $\mathbf{C}_j$
3: Compute chunk-level similarity: $\mathbf{S}_c \leftarrow \mathbf{Q}_c \mathbf{K}_c^\top \in \mathbb{R}^{N_c \times N_c}$
4: Initialize token-level similarity $\mathbf{S}_t \in \mathbb{R}^{L \times L}$ to zeros
5: **for** $j = 0$ to $N_c - 1$ **do**
6:     **for** $l = 0$ to $N_c - 1$ **do**
7:         $\mathbf{S}_t[b_j : b_{j+1}, b_l : b_{l+1}] \leftarrow \mathbf{S}_c[j, l]$
8:     **end for**
9: **end for**
10: Initialize mask $\mathbf{M} \in \{0, 1\}^{L \times L}$ to zeros
11: **for** $i = 1$ to $L$ **do**
12:     Select Top$N_b$ proceeding keys based on $\mathbf{S}_t[i, :]$
13:     Update $\mathbf{M}[i, :]$ accordingly
14: **end for**
15: **return** $\mathbf{M}$

---

**Algorithm 2** DHSA: Decode Stage

**Require:** prompt tokens $\{\mathbf{t}_i\}_{i=0}^{L-1}$; prompt chunk boundaries $\{b_j\}_{j=0}^{N_c}$ and chunks $\{\mathbf{C}_j\}_{j=0}^{N_c-1}$; current total length $L'$; previous generated tokens $\{\mathbf{t}_i\}_{i=L}^{L'-2}$; current query token $\mathbf{t}_{L'-1}$; per-query token budget $N_b$
**Ensure:** last row of the updated token-level sparsity mask $\mathbf{m}_{\text{new}}$
1: Define the updated boundaries $b_{N_c+1} \leftarrow L' - 1$ and $b_{N_c+2} \leftarrow L'$
2: Define the updated chunk $\mathbf{C}_{N_c} \leftarrow [\mathbf{t}_L, \ldots, \mathbf{t}_{L'-2}]$ and $\mathbf{C}_{N_c+1} \leftarrow [\mathbf{t}_{L'-1}]$
3: Update chunk-level keys $\mathbf{K}_c'$ for $\mathbf{C}_{N_c}$ and $\mathbf{C}_{N_c+1}$
4: Compute last row of the updated chunk-level similarity $\mathbf{s}_{c,\text{new}} \leftarrow \mathbf{q}_{L'-1}(\mathbf{K}_c')^\top \in \mathbb{R}^{N_c+2}$
5: Initialize last row of the updated token-level similarity $\mathbf{s}_{t,\text{new}} \in \mathbb{R}^{L'}$ with zeros
6: **for** $j = 0$ to $N_c + 1$ **do**
7:     $\mathbf{s}_{t,\text{new}}[b_j : b_{j+1}] \leftarrow \mathbf{s}_{c,\text{new}}[j]$
8: **end for**
9: Initialize mask $\mathbf{m}_{\text{new}} \in \{0, 1\}^{L'}$ with zeros
10: Select Top $N_b$ proceeding keys based on $\mathbf{s}_{t,\text{new}}$
11: Update $\mathbf{m}_{\text{new}}$ accordingly
12: **return** $\mathbf{m}_{\text{new}}$

---

Table 2: Performance of different methods across base models on LongBench (token density = 12.5%). All baselines are evaluated under matched sparsity settings. DuoAttention, SeerAttention, and Quest do not support Qwen2.5-3B-Instruct or gemma-2-2b-it, and are therefore omitted for those models.

| Methods | Single Doc QA | Multi Doc QA | Summ. | Few-shot Learning | Synth. | Code | Avg. |
|---|---|---|---|---|---|---|---|
| *Llama-3.1-8B-Instruct (4-bit)* | 22.0 | 10.5 | 29.4 | 68.3 | 44.0 | 22.3 | 32.7 |
| StreamingLLM | 15.0 | 6.8 | **27.7** | 62.3 | 26.0 | 22.0 | 27.0 |
| StreamingLLM w/ dilated | 15.1 | 6.7 | 27.6 | 62.3 | 26.0 | 22.1 | 27.0 |
| StreamingLLM w/ strided | 12.0 | 5.0 | 25.5 | 58.9 | 11.3 | **24.1** | 23.4 |
| MInference | 17.6 | 8.2 | 26.7 | 67.8 | 24.3 | 22.1 | 28.4 |
| Block Sparse | 16.3 | 7.4 | 22.3 | 59.7 | 44.2 | 20.5 | 27.9 |
| DuoAttention | 13.0 | 6.5 | 26.4 | 60.9 | 3.5 | 22.5 | 23.3 |
| SeerAttention | 19.9 | 10.4 | 25.6 | 68.7 | 40.1 | 19.5 | 30.8 |
| Quest | **22.4** | **10.7** | 27.1 | 60.3 | 45.2 | 21.4 | 30.9 |
| **DHSA** | 18.3 | 10.1 | 26.9 | **68.8** | **45.7** | 22.7 | **31.8** |
| *Qwen2.5-3B-Instruct* | 12.7 | 6.9 | 25.5 | 66.9 | 20.0 | 19.8 | 26.0 |
| StreamingLLM | 9.2 | 6.0 | **25.7** | 54.8 | 2.5 | 19.8 | 20.7 |
| StreamingLLM w/ dilated | 9.7 | 6.2 | 25.0 | 54.0 | 3.3 | 19.6 | 20.7 |
| StreamingLLM w/ strided | 8.1 | 7.0 | 24.7 | 47.2 | 2.1 | 18.1 | 18.8 |
| MInference | 10.1 | 5.5 | 24.1 | 57.0 | 2.5 | 19.6 | 21.1 |
| Block Sparse | 9.6 | 4.6 | 21.3 | 56.1 | 10.0 | 17.5 | 20.6 |
| **DHSA** | **11.3** | **7.9** | 24.7 | **67.2** | **14.0** | **21.6** | **25.3** |
| *gemma-2-2b-it* | 27.0 | 26.2 | 24.7 | 65.0 | 7.5 | 25.5 | 30.9 |
| StreamingLLM | 13.6 | 15.2 | 22.9 | 53.4 | 7.5 | 26.2 | 23.9 |
| StreamingLLM w/ dilated | 14.8 | 16.5 | 22.3 | 52.2 | **10.0** | **28.7** | 24.7 |
| StreamingLLM w/ strided | 14.1 | 15.1 | **23.2** | 51.2 | **10.0** | 24.4 | 23.7 |
| MInference | 17.3 | 20.8 | 22.3 | 58.0 | 5.0 | 27.5 | 26.3 |
| Block Sparse | 11.4 | 17.4 | 16.3 | 51.6 | 2.5 | 25.1 | 21.6 |
| **DHSA** | **27.6** | **24.5** | 23.1 | **64.4** | 7.5 | 25.4 | **30.3** |

## 4 EXPERIMENTS

In this section, we evaluate DHSA in terms of both effectiveness and efficiency. For effectiveness, we conduct experiments on the comprehensive long-context benchmark LongBench (Bai et al., 2023) as well as the Needle-in-a-Haystack task (Kamradt, 2023). For efficiency, we analyze kernel-level latency on both GPUs and CPUs to measure the computational benefits of DHSA.

**Implementation Details.** Our experiments use LLMs from three widely adopted open-source families: LLaMA, Qwen, and Gemma. To fit on-device applications, we select Llama-3.1-8B-Instruct (Dubey et al., 2024), Qwen2.5-3B-Instruct (Team, 2024), and gemma-2-2b-it (Gemma Team et al., 2024). We apply *4-bit quantization* for Llama-3.1-8B-Instruct, and *torch.bfloat16* precision for Qwen2.5-3B-Instruct and Gemma-2-2B-it. All GPU-related experiments are conducted on a single NVIDIA RTX 3090 GPU. For CPU experiments, we use an Intel Core 5 120U processor. To ensure stable results, all experiments use greedy decoding. Additional details are provided in Appendix D.

Our method is implemented in PyTorch with Hugging Face Transformers, leveraging either (i) the PyTorch scaled dot-product attention (SDPA) backend or (ii) FlashAttention-2 (FA2) (Dao et al., 2022). For training the boundary detector, we use Long Data Collections[1], TriviaQA (Joshi et al., 2017), and ChatQA2 (Xu et al., 2024). The configuration includes a context window size of $w = 4$ (i.e., 8 tokens per position), 8 attention heads, average pooling, an MLP hidden size of 256, and a total model size of 20 MB shared across layers and datasets. Additional details for boundary predictor training and kernel implementation are provided in Appendix B and Appendix D.

**Baselines.** In addition to dense attention, we include eight sparse attention baselines that do not require retraining the base model for a thorough comparison across a wide range of baselines: 1) StreamingLLM (Xiao et al., 2023), corresponding to the *A-shape* pattern. 2) StreamingLLM w/ dilated (Beltagy et al., 2020), which applies dilated local windows with fixed intervals. 3) StreamingLLM w/ strided (Child et al., 2019), which combines local windows with dilated attention. 4) MInference (Jiang et al., 2024), which supports *A-shape*, *Vertical-Slash*, and *Block-Sparse patterns*. 5) Block-Sparse Attention (Han, 2024), which selects tokens via blockwise similarity but lacks dynamic chunking and enhanced block representations compared to DHSA. 6) DuoAttention (Xiao et al.,

---

[1]https://huggingface.co/datasets/togethercomputer/Long-Data-Collections

Table 3: Effect of Token Density on Method Performance (LongBench).

| Methods | Single Doc QA | Multi Doc QA | Summ. | Few-shot Learning | Synth. | Code | Avg. |
|---|---|---|---|---|---|---|---|
| *Llama-3.1-8B-Instruct (4-bit)* | 22.0 | 10.5 | 29.4 | 68.3 | 44.0 | 22.3 | 32.7 |
| *Density = 6.25%* | | | | | | | |
| StreamingLLM | 13.1 | 5.3 | **26.4** | 60.4 | 20.2 | **23.9** | 25.2 |
| MInference | 14.4 | 6.9 | 20.9 | **64.4** | 20.0 | 19.7 | 25.1 |
| Block Sparse | 10.6 | 6.0 | 19.2 | 46.7 | 22.3 | 17.6 | 20.5 |
| DuoAttention | 11.5 | 5.2 | 24.8 | 59.5 | 2.7 | 21.0 | 21.9 |
| SeerAttention | 20.0 | 8.0 | 23.9 | 59.4 | 24.0 | 17.4 | 26.1 |
| Quest | **20.4** | **10.2** | 25.3 | 58.3 | 24.0 | 18.5 | 26.7 |
| **DHSA** | 18.1 | 7.2 | 21.2 | **64.4** | **36.3** | 20.2 | **27.7** |
| *Density = 12.5%* | | | | | | | |
| StreamingLLM | 15.0 | 6.8 | **27.7** | 62.3 | 26.0 | 22.0 | 27.0 |
| MInference | 17.6 | 8.2 | 26.7 | 67.8 | 24.3 | 22.1 | 28.4 |
| Block Sparse | 16.3 | 7.4 | 22.3 | 59.7 | 44.2 | 20.5 | 27.9 |
| DuoAttention | 13.0 | 6.5 | 26.4 | 60.9 | 3.5 | 22.5 | 23.3 |
| SeerAttention | 19.9 | 10.4 | 25.6 | 68.7 | 40.1 | 19.5 | 30.8 |
| Quest | **22.4** | **10.7** | 27.1 | 60.3 | 45.2 | 21.4 | 30.9 |
| **DHSA** | 18.3 | 10.1 | 26.9 | **68.8** | **45.7** | **22.7** | **31.8** |
| *Density = 25.0%* | | | | | | | |
| StreamingLLM | 20.5 | 7.7 | 28.2 | 61.6 | 36.3 | 22.8 | 29.5 |
| MInference | 21.9 | 9.5 | 28.8 | 67.5 | 35.9 | 21.5 | 31.1 |
| Block Sparse | 18.2 | 8.9 | 26.7 | 62.4 | 42.1 | 23.2 | 30.0 |
| DuoAttention | 14.5 | 8.0 | 28.1 | 62.4 | 5.0 | **23.8** | 24.8 |
| SeerAttention | 21.7 | 9.2 | 28.2 | 68.7 | 44.5 | 23.2 | 32.4 |
| Quest | 22.8 | **10.8** | 28.7 | 60.5 | 41.2 | 23.6 | 31.1 |
| **DHSA** | **24.3** | 10.2 | **28.9** | **69.4** | **52.7** | 23.7 | **34.4** |

2024), which selects retrieval heads and local heads for a sparse KV cache. 7) SeerAttention (Gao et al., 2024), which learns attention gates on top of block-sparse attention, with the key component being prediction of the block mask. 8) Quest (Tang et al., 2024), which compresses the KV cache to lower memory footprint and improve decode-time efficiency. Details of the hyperparameter settings for all baselines are presented in Appendix D.

**LongBench.** As shown in Table 2, DHSA achieves the best overall performance on LongBench compared to all baseline methods. Notably, DHSA matches the performance of the original dense attention baseline with a token density of only 12.5% (defined in Appendix C.6), while providing significant reductions in latency and memory usage. Across tasks such as QA, few-shot learning, and synthetic benchmarks, DHSA consistently outperforms existing sparse attention methods.

**Effect of Token Density.** As shown in Table 3, we investigate the impact of token density (defined in Appendix C.6) on model performance. Across all density levels, DHSA consistently outperforms baseline methods and benefits most from larger attention budgets. Notably, at 25% density, DHSA even surpasses the original dense attention baseline on tasks such as QA and synthetic tasks. Overall, DHSA scales monotonically with density and maintains stable performance across tasks.

**Needle-In-A-Haystack.** As shown in Figure 1a, our method reliably retrieves information placed at different positions across context windows ranging from 1K to 100K tokens. In contrast, baselines such as Block Sparse Attention, while effective in reducing latency, suffer a sharp performance drop once the critical information lies outside their restricted attention ranges (see Figure 13). This

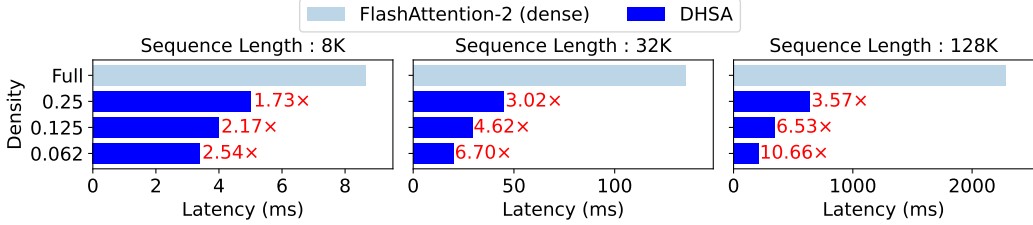

Figure 7: DHSA speedup over FlashAttention-2 at the attention kernel level during the prefill stage. Results are obtained using 4-bit Llama-3.1-8B-Instruct on an NVIDIA RTX 3090. DHSA consistently reduces latency across all evaluated sequence lengths and density settings.

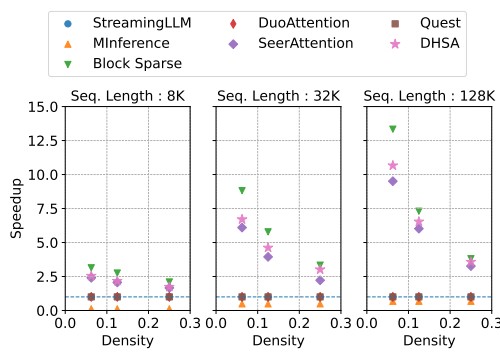

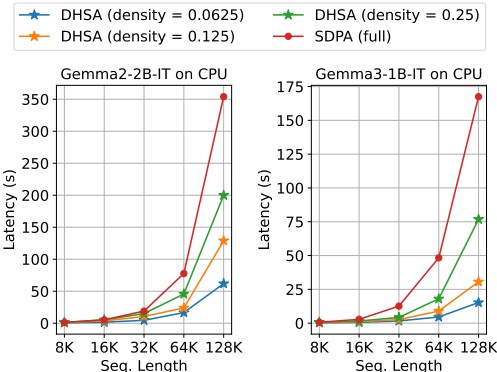

Figure 8: Kernel-level prefill speedup over Flash Attention-2 across multiple sparse attention methods. Note that StreamingLLM, DuoAttention, and Quest primarily target KV-cache compression and therefore do not accelerate prefill latency.

Figure 9: DHSA speedup over Torch.SDPA at the attention-kernel level during the prefill stage. Measurements are obtained on an Intel Core 5 120U CPU.

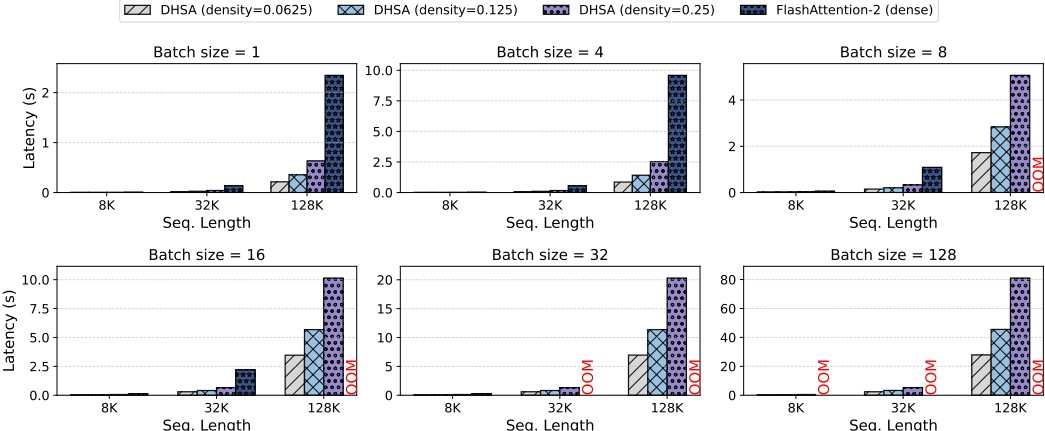

Figure 10: DHSA prefill speedup over FlashAttention-2 at the kernel level across different batch sizes. DHSA processes batches with a simple for-loop, which is faster than batched FA2 and avoids the OOM failures that FA2 often encounters. Results are measured with 4-bit Llama-3.1-8B-Instruct on an NVIDIA RTX 3090.

limitation highlights the trade-off between efficiency and accuracy in fixed or heuristic sparsity patterns, which fail to adapt to varying token distributions.

**Latency Comparison.** We first report the DHSA prefill speedup over FlashAttention-2 at the attention-kernel level (Figure 7). Across all evaluated sequence lengths and density levels, DHSA consistently reduces latency. We further compare kernel-level prefill speedup against other sparse baselines in Figure 8. DHSA delivers substantial acceleration and approaches the speedup of Block-Sparse Attention. To demonstrate that DHSA generalizes beyond GPUs, we also benchmark kernel-level prefill speedup over Torch.SDPA on an Intel Core 5 120U CPU (Figure 9). Finally, we evaluate end-to-end prefill latency for LLaMA-3.1-8B (4-bit) on a single NVIDIA RTX 3090 (Figure 1b), and present additional results on Gemma-2-2B-IT comparing dense attention and DHSA across multiple backends (Table 5). Across all settings, DHSA consistently accelerates the prefill stage while maintaining high accuracy.

Table 4: Performance of different ablation methods using Llama-3.1-8B (4-bit) on LongBench (density = 12.5%).

| Methods | Single Doc QA | Multi Doc QA | Summ. | Few-shot Learning | Synth. | Code | Avg. |
|---|---|---|---|---|---|---|---|
| DHSA | **18.3** | **10.1** | 26.9 | **68.8** | **45.7** | **22.7** | **31.8** |
| DHSA w/o robust chunk repr. | 17.1 | 8.2 | **27.0** | 67.2 | 45.0 | 21.4 | 30.7 |
| DHSA w/o dynamic chunking | 16.4 | 7.4 | 22.1 | 59.8 | 44.0 | 20.6 | 28.0 |
| DHSA w/o robust chunk repr. & dynamic chunking | 16.3 | 7.4 | 22.3 | 59.7 | 44.2 | 20.5 | 27.9 |

**Batch Inference.** Batch inference is inherently challenging for dynamic sparse attention. Rather than enforcing a shared sparsity mask across all sequences in a batch, DHSA processes each example sequentially with a lightweight for-loop. In practice, this per-example batching strategy is faster than batched FlashAttention-2 (FA2) and avoids the OOM failures that FA2 frequently encounters. All results are obtained with 4-bit Llama-3.1-8B-Instruct on an NVIDIA RTX 3090 (see Figure 10).

**Ablation Study on the Components of DHSA.** To assess the contribution of different components in DHSA, we evaluate three variants: 1) DHSA without robust chunk representation; 2) DHSA without dynamic chunking; 3) DHSA without both dynamic chunking and robust chunk representation, which reduces the method to standard block-sparse attention. The results in Table 4 show that dynamic chunking substantially improves performance by better capturing the semantic structure of tokens. Furthermore, robust chunk representation provides additional gains by preventing important tokens from being overshadowed by less relevant ones.

## 5 RELATED WORK

**Sparse Attention** The quadratic cost of attention has motivated sparse alternatives, broadly divided into *static* and *dynamic* methods. *Static* patterns such as sliding windows (Child et al., 2019), dilated/strided schemes (Beltagy et al., 2020; Ding et al., 2023), and local–global mixtures (Beltagy et al., 2020; Zaheer et al., 2020) are usually hardwired at pretraining, making them difficult to use as drop-in replacements without accuracy loss. *Dynamic* methods adapt sparsity at inference. Examples include MInference (Jiang et al., 2024), which combines pre-defined templates with kernel-aware indexing, and block-sparse attention (Han, 2024), which uses blockwise attention to select important blocks. However, these approaches still rely heavily on templates or heuristics, limiting their ability to capture semantic structure. A separate line of work integrates state-space and linear attention into structured masked attention (Gu et al., 2021; Gu and Dao, 2023; Dao and Gu, 2024).

**Long-Context LLM Inference** Long-context inference faces both high attention cost and large KV cache storage. Prefill optimizations include state-space models (Gu et al., 2021; Gu and Dao, 2023), linear attention (Sun et al., 2023), memory-based methods (Munkhdalai et al., 2024), hybrids (Lieber et al., 2024), and prompt compression (Li et al., 2023; Jiang et al., 2023), though most require retraining or add overhead. Recent work (Jiang et al., 2024; Han, 2024) instead uses predefined templates or heuristics. Decoding optimizations focus on static (Xiao et al., 2023; Han et al., 2023b) and dynamic (Zhang et al., 2023; Liu et al., 2023) KV cache dropping, and KV cache quantization (Liu et al., 2024), but these do not reduce prefill attention cost.

**On-Device LLMs** The primary challenges for on-device inference are memory footprint, compute, and limited bandwidth/power. Practical systems address these constraints by combining model compression (8/4-bit quantization (Dettmers et al., 2023), activation/KV quantization (Liu et al., 2024), pruning (Frantar and Alistarh, 2023)), capacity transfer (distillation to small models (Sanh et al., 2019; Zhang et al., 2024)), kernel-level optimizations (Flash/SDPA attention (Dao et al., 2022), operator fusion, paged attention (Kwon et al., 2023), speculative decoding (Leviathan et al., 2023)), and context management (prompt compression (Li et al., 2023; Jiang et al., 2023), cache eviction/dropping (Xiao et al., 2023; Zhang et al., 2023; Li et al., 2024)). These techniques are largely orthogonal: quantization reduces storage and compute costs, KV compression and eviction bound runtime memory, while decoding accelerations reduce token-generation latency but leave prefill's quadratic cost unaddressed. Our method is complementary, introducing input-adaptive sparsity to cut prefill attention overhead while preserving accuracy, and can be integrated as a drop-in component within existing on-device stacks.

## 6 CONCLUSION

We presented Dynamic Hierarchical Sparse Attention (DHSA), a method that integrates dynamic chunking with hierarchical sparsity prediction to efficiently focus attention on the most relevant tokens in long-context language modeling. Experiments on the Needle-in-a-Haystack Test and LongBench show that DHSA matches the accuracy of dense attention while significantly reducing latency and memory consumption. Unlike static sparsity or heuristic-based templates, DHSA adapts to input-dependent attention patterns, leading to more effective use of the attention budget and providing a practical solution for efficient long-context modeling on resource-constrained devices.

## LIMITATIONS

The primary objective of DHSA is to accelerate on-device LLM inference. While extending the maximum context length could further enhance its utility, this direction lies beyond the scope of the present work and is left for future exploration. Such an extension would not only broaden the range of applications but also provide a test of DHSA's scalability under extreme long-context scenarios. Moreover, although DHSA is fully data-driven, its performance is influenced by two key hyperparameters: the maximum number of chunks and the per-query token budget. These hyperparameters control the balance between efficiency and accuracy, forming the sparsity budget that governs the trade-off. The optimal values for these parameters depend on the specific model, task, and hardware configuration. To address this, future research could focus on developing adaptive or learned strategies for budget allocation, making DHSA more flexible and deployable across diverse environments.

## ETHICS STATEMENT

This work does not involve human subjects, personal data, or sensitive demographic information. All datasets are publicly available and used under their respective licenses. Our method aims to improve the efficiency of large language models, which can promote accessibility and sustainability. We acknowledge that LLMs may be misused for generating harmful or biased content, but our work does not specifically target such applications. No conflicts of interest or ethical concerns are associated with this research.

## REPRODUCIBILITY STATEMENT

We have made significant efforts to ensure the reproducibility of our work. The main paper and appendix provide detailed descriptions of our model architecture, training procedures, and evaluation settings. All datasets used are publicly available, and we include a complete description of data processing steps in the supplementary materials. Pseudocode and complexity analysis are provided in the paper and appendix to clarify algorithmic details.

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

# A    LLM USAGE

We used Large Language Models (LLMs) only as assistive tools for grammar refinement, readability improvements, and LaTeX formatting. They were not involved in generating research ideas, designing methods, conducting experiments, or analyzing results. All technical content and conclusions are entirely the work of the authors.

# B    BOUNDARY DETECTION

## B.1    PROBLEM FORMULATION

We formulate the chunking task as a **boundary detection** problem, where the objective is to determine whether each token position marks the *end of a chunk*. Formally, for each position $i \in [0, L-1]$, we define a boundary indicator function

$$\delta(i) = \begin{cases} 1 & \text{if } i = b_k \text{ for some } k, \\ 0 & \text{otherwise,} \end{cases}$$

indicating that the token at position $i$ is the last token of a chunk. This end-boundary prediction approach aligns naturally with the way sequences are segmented, as it allows the model to determine when a coherent segment of context has concluded. By framing the task this way, we can leverage binary classification methods to adaptively segment sequences in DHSA.

This probability, denoted as $\Pr(\delta(i) = 1)$, is predicted based on the local key representations surrounding $i$, using a boundary prediction function that takes as input two windows centered at $i$:

$$[\mathbf{k}_{i-w+1}, \cdots, \mathbf{k}_i], \quad [\mathbf{k}_{i+1}, \cdots, \mathbf{k}_{i+w}],$$

where $w$ is a hyperparameter that determines the receptive field and $\mathbf{k}_j$ denotes the key vector corresponding to token $j$.

Local context is sufficient for boundary prediction because chunk boundaries are determined by **local changes in semantic similarity**. Intuitively, if the left window (preceding tokens) and the right window (succeeding tokens) are highly similar, the two regions likely belong to the same chunk, and no boundary should be placed. Conversely, a sharp drop in similarity between these windows indicates a topic or context shift, suggesting the end of a chunk. Moreover, focusing on local context rather than the full sequence significantly reduces computation. Evaluating boundaries requires processing only tokens per position which is essential for **efficiency**.

## B.2    ARCHITECTURE

We define the boundary prediction function as a neural network. After exploring various architectures and hyperparameters, we finalize the design shown in Figure 6. This architecture was chosen because it strikes a balance between **expressiveness**, **efficiency**, and **robustness**. The encoder effectively captures contextualized token embeddings for both the left and right windows. The feature fusion module proved more stable and discriminative than using a single metric in isolation. The final MLP is shallow enough to maintain low latency but still has sufficient capacity. By keeping the receptive field limited to $2w$ tokens, the total prediction cost is linear to token length $L$.

Specifically, in the **feature fusion** module, we combined the raw context vectors $\mathbf{k}_{\text{left}}, \mathbf{k}_{\text{right}}$, absolute differences $|\mathbf{k}_{\text{left}} - \mathbf{k}_{\text{right}}|$, multiplicative interactions $\mathbf{k}_{\text{left}} \odot \mathbf{k}_{\text{right}}$, and cosine similarity $\text{sim}(\mathbf{k}_{\text{left}}, \mathbf{k}_{\text{right}})$ because each signal captures complementary aspects of boundary semantics. The raw vectors preserve local context information, absolute differences highlight directional changes between left and right spans, multiplicative interactions emphasize co-activation patterns, and cosine similarity provides a normalized measure of alignment. Together, these features make the boundary predictor more robust to scale, length, and semantic variation. In ablations, we found that using only a single similarity measure (e.g., cosine similarity) was less stable, whereas combining multiple signals yielded consistently better boundary detection.

## B.3 AUTOMATIC LABELING

To train the boundary predictor, we automatically derive ground-truth labels from attention scores, avoiding the need for manual annotation. We adopt this intermediate labelling strategy instead of end-to-end training from final task performance. End-to-end optimization is computationally intractable, as it would require differentiating through boundary indices with only sparse, delayed supervision from downstream metrics. In contrast, derived labels provide dense, local supervision, turning boundary detection into a well-defined, efficient classification problem that still captures the structural cues implicit in the model's own attention behavior.

Specifically, we analyze accumulated attention mass patterns. Tokens within a coherent span typically exhibit consistent accumulated attention profiles, meaning the distribution of attention mass over preceding tokens remains relatively stable across positions within the span. In contrast, a boundary is often marked by a sudden change in this profile, such as when the subsequent token's accumulated attention shifts sharply toward a different subset of preceding tokens.

Figure 11 illustrates our automatic labeling strategy. For each candidate position, we examine the accumulated attention mass patterns in its left window and right window, where each row in the heatmap corresponds to a token and color intensity indicates the magnitude of accumulated attention mass. In the left example, the left and right windows (both outlined in blue) exhibit highly similar attention profiles, indicating that the tokens around this position belong to the same coherent span; thus, the position is not labeled as a boundary. In the right example, the attention profiles of the left window (blue) and right window (green) differ markedly, signaling a sharp change in attention behavior. This difference suggests that the position lies at the end of a chunk and should be labeled as a boundary.

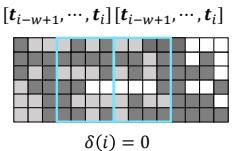 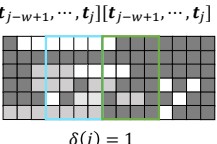

$\delta(i) = 0$ $\qquad\qquad$ $\delta(j) = 1$

(a) Non-boundary: similar left/right attention mass. (b) Boundary: different left/right attention mass.

Figure 11: Illustration of our automatic labeling strategy. The core intuition is that tokens within a coherent span tend to receive similar attention distributions. Positions where the left and right windows show similar attention distributions are labeled as non-boundary, whereas positions with sharp differences between the two windows indicate a boundary.

Formally, let the token sequence be

$$\mathbf{T} = [\mathbf{t}_0, \mathbf{t}_1, \ldots, \mathbf{t}_{L-1}]$$

with length $L$, and let $A \in \mathbb{R}^{L \times L}$ denote its attention matrix, where $A_{u,v}$ is the attention weight from token $u$ to token $v$.

For each position $i \in [0, L-1]$, we consider whether it marks the end of a chunk. To do so, we examine two local windows of size $w$ on either side of $i$, i.e., the past window: tokens $\{\mathbf{t}_{i-w+1}, \ldots, \mathbf{t}_i\}$ and the future window: tokens $\{\mathbf{t}_{i+1}, \ldots, \mathbf{t}_{i+w}\}$.

We then compute the past cumulative attention mass:

$$a_{\text{past}}(i) = \frac{1}{L-1-i-w} \sum_{u=i+w+1}^{L-1} \sum_{v=i-w+1}^{i} A_{u,v}, \tag{5}$$

and the future cumulative attention mass:

$$a_{\text{fut}}(i) = \frac{1}{L-1-i-w} \sum_{u=i+w+1}^{L-1} \sum_{v=i+1}^{i+w} A_{u,v}, \tag{6}$$

where $w = 4$ is the local window size (receptive field), and the outer sum index $u$ iterates over future tokens (beyond $i + w$) while the inner sum index $v$ iterates over the tokens in the corresponding window.

We define the attention ratio:

$$r_i = \frac{\max\left(a_{\text{fut}}(i), a_{\text{past}}(i)\right) + \varepsilon}{\min\left(a_{\text{fut}}(i), a_{\text{past}}(i)\right) + \varepsilon}, \tag{7}$$

where $\varepsilon = 0.001$ is added for numerical stability.

Given the maximum allowed number of chunks $N_c$ and a threshold $\theta_r = 1.1$, we select the top $N_c - 1$ positions $i$ whose $r_i$ exceeds $\theta_r$ (with positions 0 and $L$ always included as boundaries), with $N_c$ serving as the primary constraint in practice.

### B.4 TRAINING OF THE BOUNDARY PREDICTOR

We describe below the key implementation details of our training pipeline that are non-trivial and contribute meaningfully to performance.

**Soft Labels.** Instead of using hard labels, which are obtained by selecting the top $N_c - 1$ positions with the highest attention ratios, we adopt a soft labeling strategy. Hard labels are inherently sensitive to the chunk number constraint, i.e., the fixed number $N_c - 1$ of chunk boundaries to be selected per sequence. Under this scheme, a position with a given ratio $r$ might receive a label of 1 for one choice of $N_c$ but a label of 0 for another, even though its underlying ratio has not changed. This sensitivity can confuse the model and discard useful information about positions near the cutoff.

To address this, we convert the attention ratio $r$ into a continuous probability value using the following transformation:

$$p = \sigma\big(\alpha \cdot (\log(r + \zeta) - \beta)\big) \tag{8}$$

where $r$ is the ratio, $\zeta = 10^{-6}$ is a small constant for numerical stability, $\alpha = 2.0$ and $\beta = \lg(2.0)$ are scalar parameters controlling the slope and offset, with base $e \approx 2.71828$, and $\sigma(x) = \frac{1}{1+e^{-x}}$ is the sigmoid function.

This maps the ratio to a probability in the $[0, 1]$ range, preserving the relative ordering of positions and capturing confidence without enforcing a hard cutoff. Positions with higher ratios produce probabilities closer to 1, but those with moderately high ratios still receive meaningful supervision. This approach is in line with knowledge distillation, where soft targets have been shown to improve generalization by providing richer learning signals than binary labels.

**Loss.** Using the conventional Binary Cross-Entropy (BCE) loss for training our boundary predictor in dynamic chunking, we identify two key issues: (1) *class imbalance*, where boundary tokens are much fewer than non-boundary tokens, and (2) *varying sample difficulty*, where some boundaries are easier to detect than others. To address these, we adopt the **focal BCE loss**, which down-weights the contribution of easy examples and amplifies the focus on harder, misclassified cases. The weighting term $(1 - p_i)^\gamma$ automatically reduces the loss for well-classified positions (large $p_i$ for positives, small $p_i$ for negatives), while the fixed positive-class weight $w$ offsets the imbalance between boundary and non-boundary positions.

$$\mathcal{L}_i = (1 - p_i)^\gamma \Big[ -wy_i \log p_i - (1 - y_i) \log(1 - p_i) \Big] \tag{9}$$

where $y_i \in [0, 1]$ is the ground-truth soft label for position $i$, $z_i \in \mathbb{R}$ is the logit produced by the boundary predictor, $p_i = \sigma(z_i) = \frac{1}{1+e^{-z_i}}$ is the predicted probability, $w = 1.3$ is the fixed positive-class weight for class imbalance, and $\gamma = 2.0$ is the focal parameter controlling emphasis on hard examples.

**Data preparation.** We analyze existing long-context datasets for both training and inference. For training, we select Long Data Collections[2], trivia QA (Joshi et al., 2017) , ChatQA2 (Xu et al., 2024) that mainly focus on high-quality question answering and summarization tasks. We inspected the datasets and found that the samples are quite similar. To accelerate training, we apply sampling by selecting the first 10,000 samples from each dataset for training. For evaluation, we use the first 100 samples from the validation set of each dataset.

**Metrics.** We monitor the following metrics during training to facilitate debugging and to compare different hyperparameter configurations. We track the training loss across all layers (results shown for Gemma2-2b-it, which has 26 layers in total). For training, we evaluate precision, recall, and F1

---

[2]https://huggingface.co/datasets/togethercomputer/Long-Data-Collections

score for the positive class (where soft labels $\geq 0.5$ are treated as positive), as well as top-$K$ overlap with $K = 500$. Top-$K$ overlap is defined as the number of overlapping positions divided by $K$, and we monitor this metric because, during inference, the top positions are selected as boundaries. For validation, we compute precision, recall, F1 score, and top-$K$ overlap ($K = 500$) on the validation set.

**Acceleration.** We further accelerate training through the following strategies. (1) *Storing only boundary labels*: Instead of storing both input embeddings and boundary labels, which would consume excessive memory, we store only the labels. This allows the model to perform dynamic chunking directly from the loaded labels without recalculating them. Additionally, the labels can be generated in parallel for all training samples, significantly improving efficiency. (2) *Training all layers simultaneously*: The boundary predictor needs to be trained for all layers. Rather than performing multiple forward passes, one for each layer, we perform a single forward pass of the language model per sample to obtain input embeddings for all layers. These embeddings are then used to train the predictor for all layers simultaneously.

### B.5  INFERENCE OF THE BOUNDARY PREDICTOR

We observe that selecting positions purely based on the topK scores can lead to suboptimal boundaries due to noise and closely spaced high-score peaks. To mitigate this, we adopt Non-Maximum Suppression (NMS) (Neubeck and Van Gool, 2006), a technique widely used for eliminating redundant detections in computer vision. The core idea is to keep only the highest-scoring candidate within a local neighborhood, thereby producing well-separated, reliable boundaries.

We first obtain the boundary scores by computing the boundary predictor's output scores for all positions in the sequence, which represent the model's confidence that a given position marks a chunk end. Next, we identify candidate boundary positions by selecting all positions whose scores exceed a minimal confidence threshold, thereby pruning out low-probability positions while retaining multiple plausible candidates. The remaining candidates are then sorted in descending order of their scores, ensuring that higher-confidence positions are considered first during suppression. We subsequently apply NMS: starting with the highest-scoring candidate, we mark it as a boundary and remove any other candidates within a specified window size (e.g., 8 or 64 tokens) of this position, as they are considered overlapping or too close to be separate boundaries. After suppression, the final set consists of well-spaced local maxima that are more robust to noise and score fluctuations.

Applying NMS with a small window size (e.g., 8) yields better results than using no NMS. For tasks requiring broader context segmentation, a larger NMS window size (e.g., 64) further improves performance. In some cases, we also augment the output with explicit boundaries such as `\n`, `\n\n`, or prior information from structured prompts to better align with semantic structure.

## C  THEORETICAL ANALYSIS

In this section, we provide a theoretical analysis that explains why *Dynamic Hierarchical Sparse Attention* (DHSA) can achieve higher attention recall than fixed block-sparse attention under the same sparsity budget.

### C.1  SETUP AND NOTATION

We consider a single attention head and a single layer, and start with a single query token $i \in \{1, \ldots, L\}$. Let

$$a_{ij} \geq 0, \quad j = 1, \ldots, L$$

denote the (unnormalized) attention scores from query $i$ to keys $j$. Let $I_i^\star \subseteq \{1, \ldots, L\}$ be the *oracle important set* for query $i$ (e.g., the indices of the top-$K$ keys by $a_{ij}$), and define the corresponding binary mask row

$$M_i^\star(j) = \mathbf{1}\{j \in I_i^\star\}.$$

Given any approximate sparsity pattern $\widehat{M}_i(j) \in \{0, 1\}$ for query $i$, we measure its *per-row recall* with respect to $M_i^\star$ as

$$\text{Recall}_i(\widehat{M}) = \frac{\sum_{j=1}^{L} M_i^\star(j)\, \widehat{M}_i(j)}{\sum_{j=1}^{L} M_i^\star(j)} = \frac{|I_i^\star \cap \widehat{I}_i|}{|I_i^\star|}, \tag{10}$$

where $\widehat{I}_i = \{j : \widehat{M}_i(j) = 1\}$ is the set of keys selected by the approximate method.

We assume that all methods are constrained to use the *same token budget* per query:

$$|\widehat{I}_i| = |I_i^\star| = K. \tag{11}$$

At the matrix level, the overall recall is the average $\text{Recall}(\widehat{M}) = \frac{1}{L} \sum_i \text{Recall}_i(\widehat{M})$. It therefore suffices to compare $\text{Recall}_i$ for a fixed query $i$.

We denote by $M_i^{\text{DHSA}}$ and $M_i^{\text{block}}$ the binary masks produced by our method (DHSA) and by fixed block-sparse attention, respectively, and compare $\text{Recall}_i(M^{\text{DHSA}})$ to $\text{Recall}_i(M^{\text{block}})$ under the same budget equation 11.

## C.2 SEMANTIC SEGMENT ASSUMPTION

We formalize the intuition that important keys for a given query tend to cluster into contiguous "semantic segments" (e.g., phrases, sentences, paragraphs).

**Definition C.1** (Semantic segments). *Let $\{1, \ldots, L\}$ be partitioned into contiguous segments*

$$0 = b_0 < b_1 < \cdots < b_C = L,$$

*and define segment $c$ as the index set*

$$\mathcal{S}_c = \{b_{c-1} + 1, \ldots, b_c\}, \quad c = 1, \ldots, C.$$

*For a fixed query $i$, define the* segment mean importance

$$\mu_{ic} = \frac{1}{|\mathcal{S}_c|} \sum_{j \in \mathcal{S}_c} a_{ij}. \tag{12}$$

We assume that scores vary slowly within segments but can change substantially across segments.

**Assumption C.1** (Intra-segment homogeneity and inter-segment separation). *Define*

$$\epsilon_{\text{intra}} = \max_{i,c} \max_{j,j' \in \mathcal{S}_c} |a_{ij} - a_{ij'}|$$

*and*

$$\Delta_{\text{inter}} = \min_i \min_{c \neq c'} |\mu_{ic} - \mu_{ic'}|.$$

*We assume*

$$\epsilon_{\text{intra}} \ll \Delta_{\text{inter}}. \tag{13}$$

Assumption C.1 captures a regime where tokens within the same semantic segment have similar importance, while large changes occur when moving between different segments (e.g., across sentence or topic boundaries).

## C.3 RELEVANT VS. IRRELEVANT SEGMENTS

Fix a query $i$. Suppose:

1. All segments have equal length $m$: $|\mathcal{S}_c| = m$ for all $c$, hence $C_m = L$.
2. Each segment is either *relevant* or *irrelevant* for query $i$:
   - If segment $c$ is relevant, then all tokens in $\mathcal{S}_c$ are important: $M_i^\star(j) = 1$ for all $j \in \mathcal{S}_c$.
   - If segment $c$ is irrelevant, then all tokens in $\mathcal{S}_c$ are unimportant: $M_i^\star(j) = 0$ for all $j \in \mathcal{S}_c$.

Let $\mathcal{C}_{\text{rel}} \subseteq \{1, \ldots, C\}$ be the set of relevant segments for query $i$, with size $|\mathcal{C}_{\text{rel}}| = R$. By construction, the oracle important set is

$$I_i^\star = \bigcup_{c \in \mathcal{C}_{\text{rel}}} \mathcal{S}_c, \quad |I_i^\star| = R_m. \tag{14}$$

We consider a per-row sparsity budget equal to the number of truly important tokens:

$$K = |I_i^\star| = R_m. \tag{15}$$

Under this budget, the oracle can achieve $\text{Recall}_i(M^\star) = 1$.

### C.4 DHSA WITH DYNAMIC CHUNKING

DHSA predicts *chunk boundaries* $\hat{b}_0, \ldots, \hat{b}_{\hat{C}}$ and induced chunks

$$\hat{\mathcal{S}}_k = \{\hat{b}_{k-1} + 1, \ldots, \hat{b}_k\}, \quad k = 1, \ldots, \hat{C}.$$

For each chunk and query $i$, DHSA computes a *length-normalized chunk score*

$$\hat{\mu}_{ik} = \frac{1}{|\hat{\mathcal{S}}_k|} \sum_{j \in \hat{\mathcal{S}}_k} a_{ij}, \tag{16}$$

implemented via robust chunk representation. DHSA selects the top-$N_b$ chunks according to $\hat{\mu}_{ik}$ and then selects tokens within these chunks until the budget $K$ is filled.

To relate chunks to the true segments $\mathcal{S}_c$, we assume the boundary predictor is reasonably accurate.

**Assumption C.2** (Boundary accuracy). *For each true segment $\mathcal{S}_c$, the predicted boundaries may shift the true endpoints by at most $\delta$ tokens on each side. Concretely: each $\mathcal{S}_c$ is covered by the union of at most two predicted chunks, and the number of tokens of $\mathcal{S}_c$ that lie in chunks dominated by other segments is at most $2\delta$:*

$$|\mathcal{S}_c \setminus \hat{\mathcal{S}}_c| \leq 2\delta, \tag{17}$$

*for some appropriate correspondence between true segments and predicted chunks.*

Combined with Assumption C.1, this implies that chunk means preserve the ranking of segment means.

**Lemma C.1** (Chunk score stability). *Under Assumptions C.1 and C.2, each predicted chunk $\hat{\mathcal{S}}_k$ overlaps primarily with some true segment $\mathcal{S}_c$, and its mean $\hat{\mu}_{ik}$ satisfies*

$$|\hat{\mu}_{ik} - \mu_{ic}| \leq O(\epsilon_{\text{intra}}) + O\left(\frac{\delta}{m} \Delta_{\text{inter}}\right).$$

*In particular, if $\epsilon_{\text{intra}}$ and $\delta/m$ are sufficiently small, then the ranking of chunk means $\hat{\mu}_{ik}$ is identical to the ranking of segment means $\mu_{ic}$ for query $i$.*

*Proof sketch.* Each chunk $\hat{\mathcal{S}}_k$ is a mixture of tokens from at most two adjacent true segments (by Assumption C.2), and the fraction of misassigned tokens from neighboring segments is bounded by $\delta/m$. The mean $\hat{\mu}_{ik}$ is therefore a convex combination of the means of at most two true segments plus intra-segment noise bounded by $\epsilon_{\text{intra}}$. Using equation 13 to compare this convex combination to the dominant segment mean $\mu_{ic}$ yields the desired bound and ranking preservation. $\square$

We can now bound the recall of DHSA in the above model.

**Proposition 1** (Near-oracle recall of DHSA). *Consider the model above with segment length $m$, relevant segments $\mathcal{C}_{\text{rel}}$, and budget $K = R_m$ as in equation 15. Suppose Assumptions C.1 and C.2 hold and the ranking of chunk means matches the ranking of segment means (Lemma C.1). Then there exists a choice of $N_b$ and token-level selection within the chosen chunks such that DHSA's per-row recall satisfies*

$$\text{Recall}_i(M^{\text{DHSA}}) \geq 1 - \frac{2\delta}{m}. \tag{18}$$

*Proof.* Since chunk ranking matches segment ranking, DHSA can select exactly the chunks corresponding to all relevant segments $\mathcal{S}_c, c \in \mathcal{C}_{\mathrm{rel}}$, plus possibly some chunks corresponding to irrelevant segments if needed to fill the token budget.

By Assumption C.2, for each relevant segment $\mathcal{S}_c$, at most $2\delta$ tokens lie outside the dominant chunk and may be dropped in the upsampling step. Thus the total number of important tokens missed by DHSA is at most

$$|I_i^\star \setminus I_i^{\mathrm{DHSA}}| \leq 2\delta \cdot R.$$

Using $|I_i^\star| = R_m$ from equation 14, the recall equation 10 is

$$\mathrm{Recall}_i(M^{\mathrm{DHSA}}) = 1 - \frac{|I_i^\star \setminus I_i^{\mathrm{DHSA}}|}{|I_i^\star|} \geq 1 - \frac{2\delta R}{R_m} = 1 - \frac{2\delta}{m},$$

which proves equation 18. $\square$

Thus, when $\delta \ll m$, DHSA achieves recall close to 1, i.e., nearly matches the oracle top-$K$ selector under the same sparsity budget.

## C.5 BLOCK-SPARSE ATTENTION

We now consider a fixed block-sparse pattern with block size $B$ that partitions the sequence into blocks

$$0 = q_0 < q_1 < \cdots < q_Q = L, \quad q_r = rB,$$

and

$$\mathcal{B}_r = \{q_{r-1} + 1, \ldots, q_r\}, \quad |\mathcal{B}_r| = B.$$

A block-sparse mask selects a subset of blocks for each query and allows attention only within those blocks. We assume that selecting a block contributes all $B$ of its tokens to the budget.

We consider a worst-case but realistic alignment between segments and blocks: each relevant segment of length $m \leq B$ is split across two blocks, each containing roughly $m/2$ important tokens and $B - m/2$ unimportant tokens.

Under the same budget $K = R_m$, we show that block-sparse recall can be substantially smaller than DHSA.

**Proposition 2** (Upper bound on block-sparse recall). *With the worst-case alignment of segments and blocks as described above, block size $B \geq m$, and budget $K = R_m$, the per-row recall of any fixed block-sparse pattern satisfies*

$$\mathrm{Recall}_i(M^{\mathrm{block}}) \leq \frac{m}{2B}. \tag{19}$$

*Proof.* Each selected block $\mathcal{B}_r$ contributes $B$ tokens to the selected set $\widehat{I}_i$. Under budget $K$, the maximum number of blocks that can be selected is

$$S \leq \frac{K}{B} = \frac{R_m}{B}.$$

In the worst-case alignment, each relevant segment $\mathcal{S}_c$ is split across two blocks, each containing at most $m/2$ important tokens. Thus, a single block contributes at most $m/2$ important tokens.

Hence, the total number of important tokens captured by any block-sparse pattern is upper bounded by

$$|I_i^\star \cap I_i^{\mathrm{block}}| \leq S \cdot \frac{m}{2} \leq \frac{R_m}{B} \cdot \frac{m}{2} = \frac{R_m^2}{2B}.$$

Dividing by $|I_i^\star| = R_m$ yields

$$\mathrm{Recall}_i(M^{\mathrm{block}}) = \frac{|I_i^\star \cap I_i^{\mathrm{block}}|}{|I_i^\star|} \leq \frac{R_m^2}{2B} \cdot \frac{1}{R_m} = \frac{m}{2B},$$

which proves equation 19. $\square$

When blocks are much larger than the true semantic segments ($B \gg m$), the bound equation 19 implies that block-sparse recall under a fixed budget is at most $O(m/B)$, i.e., a small fraction of the oracle mass.

### C.6 COMPARISON AND DISCUSSION

Combining Propositions 1 and 2, we obtain,

$$\text{Recall}_i(M^{\text{DHSA}}) \; \geq \; 1 - \frac{2\delta}{m}, \qquad \text{Recall}_i(M^{\text{block}}) \; \leq \; \frac{m}{2B}.$$

For realistic regimes where $m \ll B$ (semantic segments shorter than fixed blocks) and $\delta \ll m$ (accurate boundary prediction), we have

$$1 - \frac{2\delta}{m} \; \gg \; \frac{m}{2B},$$

i.e., DHSA achieves near-oracle recall, while any fixed block-sparse pattern under the same token budget can only capture an $O(m/B)$ fraction of the truly important keys in the worst case.

This analysis formalizes the intuitive advantage of *dynamic, semantics-aligned chunking*: by adapting chunk boundaries to semantic structure and using length-normalized chunk scores, DHSA selects units that closely match the true important regions in the attention map, whereas fixed block grids are oblivious to semantics and must include many uninformative tokens whenever segments are misaligned with block boundaries. Empirically, this prediction is consistent with our attention recall vs. oracle Top-$K$ results in Figure 12, where DHSA closely tracks the oracle while block-sparse baselines exhibit recall gaps under the same sparsity budget.

**Cost analysis.** We analyze the computational cost of **DHSA** as follows. Per layer, DHSA adds a linear $\mathcal{O}(L)$ pass for boundary prediction, forms chunk representations in $\mathcal{O}(L)$, computes chunk–chunk similarity in $\mathcal{O}(N_c^2)$ for $N_c$ chunks, and then performs token-level selection with a per-query budget $N_b$ in $\mathcal{O}(L\,N_b)$ without materializing a dense $L \times L$ matrix. We define the *token density* as $N_b/L$. In practice $N_b \ll L$ and $N_c \ll L$ (both are bounded by user budgets), so the dominant terms are $\mathcal{O}(L\,N_b)$, yielding near-linear scaling in $L$ for fixed $N_b$ and $N_c$. For comparison, block-sparse attention exhibits a complexity of $\mathcal{O}(L\,N_b) + \mathcal{O}(N_c^2)$, which is asymptotically similar to DHSA.

## D IMPLEMENTATION DETAILS

**Base models.** Our experiments use LLMs from three widely adopted open-source families: LLaMA, Qwen and Gemma. To fit on-device applications, we select Llama-3.1-8B-Instruct (Dubey et al., 2024), Qwen2.5-3B-Instruct (Team, 2024), and gemma-2-2b-it (Gemma Team et al., 2024). The maximum context lengths are 128K for Llama-3.1-8B-Instruct, 32K for Qwen2.5-3B-Instruct, and 8K for gemma-2-2b-it. We apply *4-bit quantization* for Llama-3.1-8B-Instruct, and *torch.bfloat16* precision for Qwen2.5-3B-Instruct and Gemma-2-2B-it. In Llama-3.1-8B-Instruct and Qwen2.5-3B-Instruct, all layers employ global attention, whereas in gemma-2-2b-it, global attention is applied in every other layer with a sliding window of 4,096 tokens.

**Environment.** All GPU-related experiments are conducted on a single NVIDIA RTX 3090 GPU (24 GB) running Ubuntu 22.04.4 LTS. The software environment includes Python 3.12, CUDA 12.4, PyTorch 2.5.1+cu124, and Transformers 4.52.3. For CPU experiments, we use an Intel Core 5 120U processor (10 cores, 12 threads, max frequency 1.40 GHz) running Windows 11. The software environment includes Python 3.11 and PyTorch 2.9.1+cpu, and Transformers 4.52.3.

**Kernel implementation.** We provide two kernel implementations tailored to different deployment scenarios. The PyTorch SDPA-based implementation offers more flexible, token-level selection and works across a wide range of models (e.g., Gemma 2/3) and environments, including CPU. In contrast, the FlashAttention-2 (block-sparse attention) implementation is more efficient but less flexible, as it operates at a fixed block granularity (block size 128) and does not support certain models or hardware settings. Taken together, these two implementations complement each other in practice.

**1) Built on PyTorch SDPA.** This kernel is implemented in PyTorch with Hugging Face Transformers, leveraging PyTorch's scaled dot-product attention (SDPA) backend. Instead of computing dense attention via a full similarity matrix, we split the computation into chunks, which substantially reduces latency and memory usage for very long contexts. At each Transformer layer, the boundary predictor uses key states with positional embeddings to predict boundary indices $\mathcal{B}$, which partition

the token sequence into variable-length chunks. For each chunk, we employ prefix-sum aggregation to efficiently compute the sum of token queries and keys, and obtain the chunk representation by dividing this sum by the chunk length and normalizing with $\sqrt{|\mathbf{C}|}$, where $|\mathbf{C}|$ denotes the chunk size.

Given the chunk queries $\mathbf{Q}_c$ and keys $\mathbf{K}_c$, we compute their dot product to form the chunk-level similarity matrix $\mathbf{S}_c = \mathbf{Q}_c \mathbf{K}_c^\top$, with time complexity $O(N_c^2)$, where $N_c$ is the number of chunks. This similarity is then expanded to the token level by mapping each entry $(\mathbf{S}_c)_{l,k}$ to the corresponding submatrix $(\mathbf{S}_t)_{[b_l:b_{l+1}],[b_k:b_{k+1}]}$ according to the chunk boundaries. For each **query chunk**, we then perform token-level Top-$N_b$ selection to identify a compact set of $N_b$ key tokens that satisfy causal constraints. The kernel **gathers the corresponding key/value tokens into dense tiles** and constructs an exact causal mask from absolute positions. Note that, since chunk sizes are variable, we cannot safely perform selection purely at the chunk level. Chunk-level Top-$K$ would admit entire chunks and can overflow the token budget. In contrast, DHSA's token-level Top-$N_b$ selection guarantees the compute cap while still leveraging chunk routing.

**2) Built on FlashAttention-2.** Our method can also utilize Block Sparse Attention (Han, 2024) by predicting a block-sparse mask with minimal modification. When selecting chunk boundaries based on predicted scores, we restrict candidate boundaries to block boundaries, i.e., the chunk boundary can only lie in $\{0, B, 2B, \dots\}$, where $B$ is the block size (128). When computing chunk-level similarity, we no longer upsample to the token level; instead, we upsample to the block level and perform Top-$K$ selection with $K = \lfloor N_b/B \rfloor$. Once the block-sparse mask is obtained, we directly invoke Block Sparse Attention to compute attention efficiently.

**Additional optimizations for memory footprint.** For very long sequences (e.g., length $> 64$K tokens), we avoid materializing large matrices by using tiled computation and promptly releasing intermediate buffers, thereby keeping the memory footprint bounded. To further reduce memory overhead, we also tile the MLP computation, enabling efficient execution on extremely long inputs. We share boundary predictions and sparsity masks across layers to avoid redundant recomputation in deeper layers. Finally, attention is computed only over the selected tokens using our kernel, reducing the complexity to $O(L \cdot N_b)$, scaling with the selected region rather than the full quadratic cost $O(L^2)$, while preserving the semantics of causal attention.

**Long-generation tasks.** For long-generation tasks, we design DHSA with an **online** boundary policy that decides where to close chunks as new tokens are generated. Concretely, the boundary predictor operates with a small $w$-token look-ahead: once token $i + w$ has been produced, it examines the local key windows $[k_{i-w+1}, \dots, k_i]$ and $[k_{i+1}, \dots, k_{i+w}]$ (the same architecture as in Section 3.2) and, if the predicted boundary probability at position (i) exceeds the threshold (after non-maximum suppression), it finalizes a chunk ending at (i) and starts a new one at $i + 1$. This streaming-style policy only delays boundary decisions by at most $w$ tokens and requires no access to future context beyond the small local window. Empirically, on our evaluated tasks the generated continuations are short (non-summarization typically <128 tokens; summarization typically <512), so in our main results we adopt a simplified variant (Algorithm 2) that treats the entire generated prefix as a single chunk, and we do not observe any degradation in accuracy.

**Baselines.** For a fair comparison, we use the same token density $N_b/L$ (6.25%, 12.5%, 25%) for all methods. For each baseline, we follow the hyperparameter choices recommended in the original papers or official implementations. To ensure stable results, all experiments use greedy decoding.

Specifically, the hyperparameters of each baseline are configured as follows:

1. **StreamingLLM** (Xiao et al., 2023), corresponding to the *A-shape* pattern. We allocate 20% of preserved keys as global tokens and 80% as local windows;

2. **StreamingLLM w/ dilated** (Beltagy et al., 2020), which applies dilated local windows with fixed intervals. We use 20% global tokens and 80% dilated windows with an interval of 1;

3. **StreamingLLM w/ strided** (Child et al., 2019), which combines local windows with dilated attention. We use 20% global tokens, 40% local windows, and 40% dilated windows with an interval of 1;

4. **MInference** (Jiang et al., 2024), which supports *A-shape*, *Vertical-Slash*, and *Block-Sparse patterns*. For clarity, we adopt the Vertical-Slash pattern, using 50% vertical-line tokens and 50% slash-line tokens with $last\_q$ equals to 64;

5. **Block-Sparse Attention** (Han, 2024), which selects tokens via blockwise similarity but lacks dynamic chunking and enhanced block representations compared to DHSA. We use a block size of 128 in all experiments.

6. **DuoAttention** (Xiao et al., 2024), which selects retrieval heads and streaming heads for a sparse KV cache. The head mask is learned during training, and we adjust the sparsity ratio in our experiments to match the desired density.

7. **SeerAttention** (Gao et al., 2024), which learns attention gates on top of block-sparse attention, with the key component being prediction of the block mask. Since it does not support a fixed token budget for sparse prefill, we directly set the non-zero ratio to match the target density.

8. **Quest** (Tang et al., 2024), which compresses the KV cache to lower memory footprint and improve decode-time efficiency. Given a token budget, it restricts each query to attend to that number of tokens during decoding. We set the token budget to correspond to the same effective density as in our setting.

For StreamingLLM and its variants, MInference, Block-Sparse Attention, and Quest, we configure the patterns so that each query selects approximately the same number of tokens implied by the token density (up to small discrepancies due to block- or window-based selection). For SeerAttention, the non-zero ratio is chosen to match the same global density, though the exact number of tokens per query can vary. DuoAttention effectively uses more tokens than other methods, since its streaming heads still attend to attention sinks and a fixed set of recent tokens. Note that for DuoAttention, SeerAttention, and Quest, we conduct experiments only on models they officially support (e.g., Llama-3.1-8B-Instruct) to ensure a fair comparison.

**Needle-In-A-Haystack.** We evaluated different baselines, starting with an assessment of the models' long-context processing capabilities through the needle-in-a-haystack test (Cai et al., 2024). This benchmark evaluates a model's ability to locate a target sentence (the needle) within a long context and is widely used for long-context language modeling. Our setup used context lengths ranging from 1K to 64K tokens and depth ranges from 0% to 100% (interval of 10%). The prompt format was: `<|im_start|> This is a very long story book: <book> {context} </book>. Based on the content of the book, Question: {retrieval_question} Answer:` with the needle sentence "The best thing to do in San Francisco is eat a sandwich and sit in Dolores Park on a sunny day." and the corresponding retrieval question "The best thing to do in San Francisco is:". We used ROUGE as the evaluation metric, and visualized results with green indicating correct and red indicating incorrect predictions.

**LongBench.** LongBench (Bai et al., 2023) comprises 16 datasets designed to evaluate different aspects of long-context processing, including single-document QA, multi-document QA, summarization, few-shot learning, synthetic tasks, and code completion. Each task consists of multiple datasets: single-document QA includes NarrativeQA, Qasper, and MultiFieldQA_en; multi-document QA includes HotpotQA, 2WikiMultihopQA, and MuSiQue; summarization includes GovReport, QMSum, and MultiNews; few-shot learning includes TREC, TriviaQA, and SAMSum; synthetic tasks include PassageCount and PassageRetrieval_en; and code completion includes LCC and RepoBench-P. Average dataset lengths range from 1.2K to 18K tokens, and performance is measured using task-specific metrics (F1, ROUGE-L, accuracy, and edit similarity). We report the average score per task across its datasets. Since each model has a maximum context length $C_{max}$, for inputs exceeding this length we retain the first 1K tokens and the last $(C_{max} - 1)$ K tokens. To fit *Llama-3.1-8B-Instruct* (4-bit) on a single NVIDIA RTX 3090 GPU (24 GB), we set its $C_{max}$ to 48K.

# E    ADDITIONAL RESULTS

**Overlap with Top-K Attention**    To assess how well different sparse attention patterns preserve important information, we compute the overlap between each method's retained keys and the Top-K attention targets over the last 4K tokens of a 32K-context sequence. We extract attention scores from all heads and layers and report the average overlap to obtain a stable, model-wide estimate. Experiments are conducted using Llama-3.1-8B (4-bit) on the Needle-in-a-Haystack task. As shown in Figure 12, DHSA consistently achieves the highest overlap across all retention budgets, while

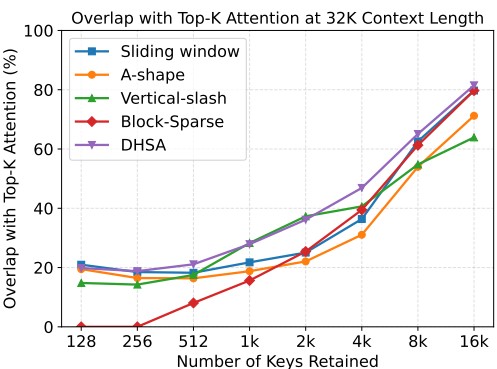

Figure 12: Overlap with Top-K attention as a function of the number of retained keys at 32K context length.

Table 5: Comparison of end-to-end prefill latency and memory usage for gemma2-2b-it (number of keys retained = 2K). FlashAttention-2 is omitted as it does not support this model. Results are measured on an NVIDIA RTX 3090.

| Context | Attn. Implem. | Method | Latency (s) | Memory (GB) |
|---|---|---|---|---|
| 16K | Eager | Dense | - | OOM |
| | | DHSA | 2.18 | 9.69 |
| | SDPA | Dense | 3.37 | **8.38** |
| | | DHSA | **1.98** | 9.69 |
| 32K | Eager | Dense | - | OOM |
| | | DHSA | 4.51 | 16.99 |
| | SDPA | Dense | 10.97 | **15.18** |
| | | DHSA | **4.13** | 16.99 |

other sparse patterns recover fewer important tokens. Overall, DHSA allocates its sparsity budget more effectively, providing the closest approximation to dense attention in long-context retrieval.

**Needle-In-A-Haystack.** As shown in Figure 1a, our method reliably retrieves information placed at different positions across context windows ranging from 1K to 100K tokens. We further evaluate several baselines, including StreamingLLM (Xiao et al., 2023), StreamingLLM w/ dilated (Beltagy et al., 2020), StreamingLLM w/ strided (Beltagy et al., 2020), MInference (Jiang et al., 2024) and Block Sparse Attention (Han, 2024). For all methods, the context length ranges from 1K to 64K tokens, and the token density (defined as the number of preserved keys divided by the context length) is fixed at 6.25%. For the 1K-token case, we preserve 512 keys, since 64 tokens (6.25% of 1024) are insufficient to maintain retrieval reliability.

From Figure 13, we observe that baselines such as StreamingLLM and Block Sparse Attention, while effective in reducing latency, suffer a sharp performance drop once the critical information lies outside their restricted attention ranges. This limitation highlights the trade-off between efficiency and accuracy in fixed or heuristic sparsity patterns, which fail to adapt to varying token distributions. In contrast, our method dynamically adjusts attention allocation, enabling it to maintain robust retrieval performance across diverse positions and context lengths while still achieving computational efficiency.

**Ablation study.** To assess the contribution of different components in DHSA, we evaluate three variants: 1) DHSA without robust chunk representation; 2) DHSA without dynamic chunking; 3) DHSA without both dynamic chunking and robust chunk representation, which reduces the method to standard block-sparse attention. The results in Table 4 demonstrate that dynamic chunking plays a critical role in performance, as it adaptively partitions sequences based on semantic boundaries rather than relying on fixed templates. This flexibility allows the model to better capture long-range dependencies and context-specific structures, leading to consistent improvements across tasks. In addition, robust chunk representation yields further gains by normalizing and refining block-level embeddings, thereby reducing the risk that important tokens are diluted by surrounding less informative ones. Together, these components enable DHSA to strike a more effective balance between sparsity and expressiveness, outperforming static sparse baselines and narrowing the gap to dense attention while maintaining efficiency.

Recall that our robust chunk representation for chunk $\mathbf{C}_k$ of length $L_k = b_{k+1} - b_k$ is defined as

$$\mathbf{q}_{\mathbf{C}_k} = \frac{\sqrt{L_k}}{L_k} \sum_{i=b_k}^{b_{k+1}-1} \mathbf{q}_i, \qquad \mathbf{k}_{\mathbf{C}_k} = \frac{\sqrt{L_k}}{L_k} \sum_{i=b_k}^{b_{k+1}-1} \mathbf{k}_i. \qquad (20)$$

Equivalently, $\mathbf{q}_{\mathbf{C}_k} = \sqrt{L_k} \bar{\mathbf{q}}_{\mathbf{C}_k}$, where $\bar{\mathbf{q}}_{\mathbf{C}_k} = \frac{1}{L_k} \sum_{i=b_k}^{b_{k+1}-1} \mathbf{q}_i$ is the average of token queries in the chunk. When dynamic chunking is removed, the boundaries $\mathcal{B}$ are fixed to uniform blocks of size $B$, i.e., $b_k = kB$ and each chunk $\mathbf{C}_k$ spans a contiguous block $[kB, (k+1)B)$. In this case, the chunk-level similarity matrix $\mathbf{S}_c = \mathbf{Q}_c \mathbf{K}_c^\top$ is proportional (up to a constant factor $B$) to the

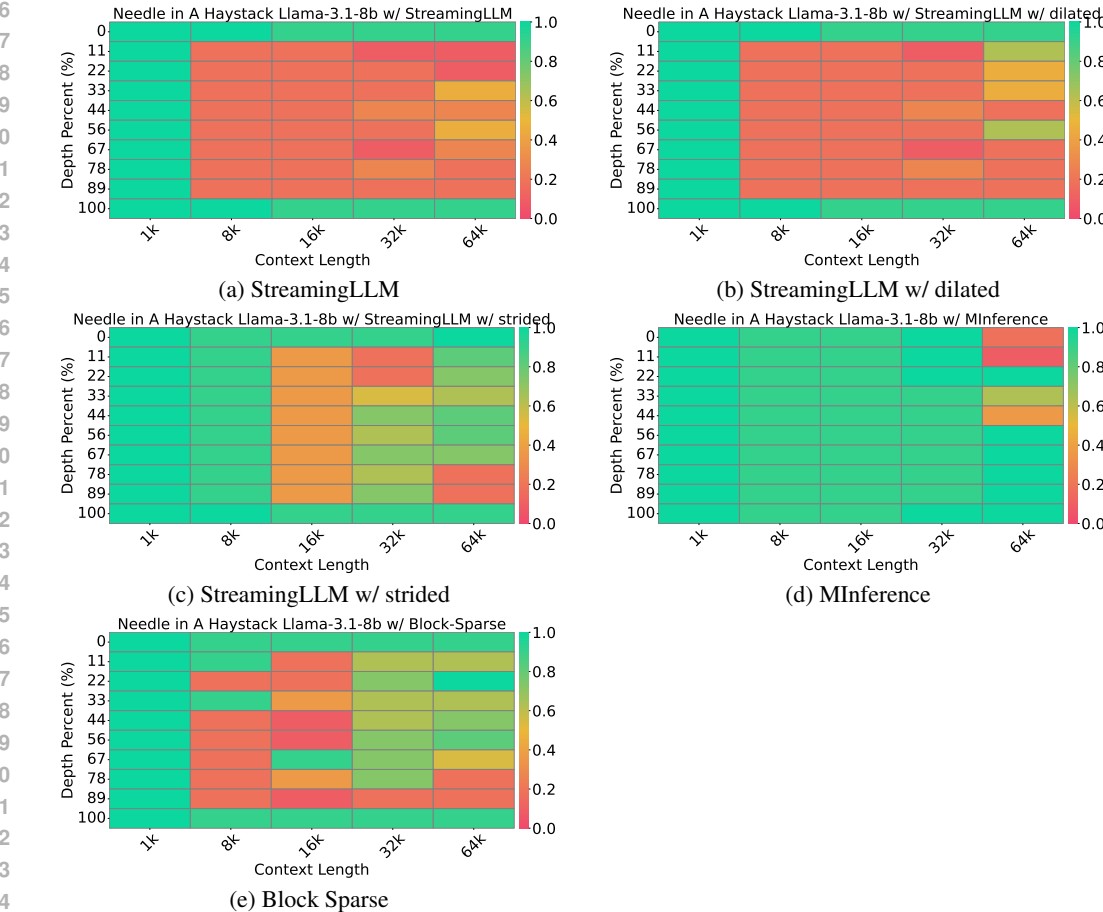

Figure 13: Needle In A Haystack results on Llama-3.1-8B (4-bit) with a token density of 6.25%.

blockwise similarity used in standard block-sparse attention with block size $B$. Since this factor is uniform across all block pairs, it does not affect which blocks are selected under a fixed sparsity budget, and the variant without dynamic chunking degenerates to a conventional block-sparse pattern.

