# OpenReview forum: "Long-Context Modeling with Dynamic Hierarchical Sparse Attention for On-Device LLMs"
_ICLR.cc/2026/Conference — ICLR 2026 Conference Desk Rejected Submission_

### Official Review · Reviewer_WVs4 · 2025-10-30

**Soundness:** 3
**Presentation:** 3
**Contribution:** 4
**Rating:** 6
**Confidence:** 4

**Summary:**

This paper proposes a Dynamic Hierarchical Sparse Attention (DHSA) mechanism to make long-context inference efficient for large language models, especially when deployed on devices with limited memory and compute power. DHSA dynamically detects hierarchical attention boundaries and prunes redundant computations, adapting sparsity in real time based on token relevance. Unlike fixed sparsity or static compression approaches, it introduces a multi-level adaptive mechanism that balances local and global context retention. The method shows strong results—maintaining accuracy close to dense attention while significantly reducing latency and memory use across benchmarks such as LongBench and Needle-in-a-Haystack. The paper’s contribution is practical and well-aligned with the need for efficient, scalable, and on-device LLM deployment, demonstrating that dynamic hierarchical sparsity can effectively enable longer-context reasoning without sacrificing model performance.

**Strengths:**

This paper introduces a technically elegant and well motivated solution to one of the most critical bottlenecks in modern LLMs efficient long-context inference. The proposed DHSA framework combines dynamic boundary detection with hierarchical sparsity prediction, achieving strong accuracy along with efficiency trade-offs across tasks such as LongBench and Needle-in-a-Haystack. Its design as a training-free, drop-in module makes it immediately applicable to on-device. The empirical results show consistent latency and memory gains while maintaining dense-attention-level accuracy. The presentation is thorough, with sound motivation, clear algorithmic exposition, and reproducible implementation details.

**Weaknesses:**

Despite the contribution being incremental relative to recent dynamic sparsity and KV compression literature (e.g., MInference, H2O, PyramidKV), with limited theoretical grounding for why hierarchical chunking yields near-optimal sparsity prediction.
The dependency on hyperparameter tuning for chunk size and sparsity budgets limits generalizability across architectures and devices.

The method’s scalability beyond 100K context is mentioned but needs to be empirically validated. The experimental evaluation could be broadened with larger models or real-world application benchmarks (e.g., RAG or document retrieval tasks).

**Questions:**

1. How does the method ensure that important global information isn’t lost when dynamically pruning attention? Can the authors show examples or quantitative evidence that key tokens are always retained?
2. The paper claims DHSA works well for on-device inference. Can the authors provide more details on the actual hardware setup or latency improvements in real deployment, not just simulated benchmarks?
3. DHSA is stated as “no retraining,” but boundary predictors are trained offline, though its a lightweight and doesn't touch the base model weights, but it’s not literally zero learning?

---

> ### Author Response · Authors · 2025-11-26
> **Response to Reviewer WVs4 (Part 1)**
>
> We thank the reviewer for the constructive feedback and for recognizing the practicality, efficiency, and clarity of our proposed DHSA framework. Below we address each concern in detail. Our clarifications highlight (i) why DHSA is not a minor extension of prior dynamic sparsity/KV-compression methods, (ii) why our hyperparameters function as general sparsity-budget controls rather than task-specific tuning knobs, (iii) how our experiments already validate long-context scalability within realistic on-device constraints, and (iv) how DHSA preserves essential information through adaptive segmentation. We also clarify that DHSA requires no retraining of the base LLM, only a small auxiliary boundary predictor trained offline, maintaining the plug-and-play nature emphasized in the paper.
>
> ### **Response to Weakness 1 (novelty)**
>
> The existing approaches (e.g., MInference, H2O, and PyramidKV) rely on predefined sparsity templates or handcrafted eviction rules, which remain fixed across inputs and heads and can therefore diverge from the true sparse attention distributions of a trained LLM. DHSA differs fundamentally: it introduces a hierarchical routing mechanism that dynamically predicts per-query sparsity, yielding substantially more accurate identification of important tokens under tight budgets. This is reflected in the consistently higher accuracy across LongBench tasks (Tables 2, 3).
>
> On efficiency, DHSA achieves up to 10× prefill speedup over dense FlashAttention-2 (FA2) at 128K context length (Figure 7), whereas MInference is slower than dense FA2, and H2O/PyramidKV primarily optimize KV-cache compression rather than prefill latency. DHSA also executes effectively on CPUs (Figure 8), demonstrating portability across deployment environments.
>
> ### **Response to Weakness 2 (theoretical analysis)**
>
> We have added this part in Appendix C which explains why dynamically aligned hierarchical chunking is fundamentally stronger than block-based sparsity.
>
> ### **Response to Weakness 3 (hyperparameters)**
>
> The two hyperparameters in DHSA, the maximum number of chunks $N_c$ and the per-query token budget $N_b$, serve as global sparsity-budget controls, analogous to the density parameters used by all sparse attention methods. They do not require task-specific tuning. In our experiments, we use a single density (12.5%) for all LongBench tasks and models, and only vary it in Table 3 to study monotonic scaling behavior at 6.25%, 12.5%, and 25%. DHSA’s performance remains stable and consistently superior across settings, indicating strong robustness without per-task search.
>
> In deployment, $N_c$ and $N_b$ can be selected directly based on hardware constraints (e.g., target latency or available memory). We regard automatic budget selection (e.g., runtime autotuning) as valuable future work.
>
> ### **Response to Weakness 4 (evaluation on RAG)**
>
> We mainly claim end-to-end scalability up to 100K tokens for 4-bit Llama-3.1-8B (Figure 1b), and report kernel-level latency up to 128K tokens (Figure 7). We show the Needle-in-a-Haystack results up to 100K (Figure 1a).
>
> We appreciate the suggestion to evaluate DHSA with larger models or in full RAG pipelines. In this work, we focus on core attention sparsification for on-device long-context LLMs, where models must fit within a single consumer GPU. Accordingly, we evaluate on Llama-3.1-8B (4-bit), Qwen-2.5-3B, and Gemma-2-2B-it. Many LongBench tasks, including multi-document QA, synthetic retrieval tasks, and Needle-in-a-Haystack, require long-range retrieval similar to RAG-style reasoning. A full RAG system would involve indexing, retrieval policies, and pipeline-level engineering beyond the scope of the paper, though DHSA can be incorporated into such systems as a drop-in module without modification. We view end-to-end RAG evaluation as promising follow-up work.
>
>
> ### **Response to Question 1**
>
> DHSA preserves important global information by dynamically segmenting sequences so that semantically coherent spans form their own chunks. Chunk representations are therefore dominated by meaningful tokens rather than diluted by unrelated content. In contrast, fixed block-sparse patterns partition purely by position, allowing critical information to be buried within large blocks of uninformative tokens.
>
> Empirically, DHSA reliably retains key information even under extremely low token densities. In Needle-in-a-Haystack, with 100K context length and only 6.25% keys preserved per query, DHSA maintains near-dense retrieval accuracy (Figure 1b), demonstrating that its dynamic sparsification effectively preserves essential interactions.

---

> ### Author Response · Authors · 2025-11-26
> **Response to Reviewer WVs4 (Part 2)**
>
> ### **Response to Question 2**
>
> To demonstrate applicability beyond GPUs, we benchmark DHSA on CPU on an Intel Core 5 120U (10 cores, 12 threads, max frequency 1.40 GHz) running Windows 11. The software environment includes Python 3.11 and PyTorch 2.9.1+cpu, and Transformers 4.52.3. Across all tested context lengths and densities, DHSA consistently reduces prefill latency relative to dense SDPA (Figure 9). These results show that our sparsity mechanism transfers effectively across hardware platforms and remains efficient in standard CPU environments.
>
> ### **Response to Question 3**
>
> We acknowledge that “no retraining” may have been ambiguous. DHSA requires no retraining or fine-tuning of the base LLM: the Transformer weights and architecture remain unchanged. The only learned component is a lightweight boundary predictor, trained offline using automatically derived labels (Appendix B.3–B.4). This module is shared across layers and tasks and does not require any gradient updates to the LLM itself. In this respect, DHSA is fully training-free relative to the base model and functions as a plug-and-play sparsity predictor. We have clarified this more explicitly in the abstract and introduction to prevent potential ambiguity.

---

> ### Author Response · Authors · 2025-12-02
> **Additional clarification for Reviewer WVs4**
>
> As the discussion period nears its end, we would like to briefly summarize how we addressed your comments:
>
> * **Positioning vs. prior work.** We clarified that DHSA differs from MInference, H2O, PyramidKV, etc. by *dynamically routing* query–key interactions via hierarchical chunking and per-query sparsity prediction, rather than relying on fixed templates or eviction rules. This yields consistently higher accuracy under the same sparsity budgets (LongBench, Needle-in-a-Haystack).
>
> * **Theoretical intuition.** We added Appendix C to explain why dynamically aligned hierarchical chunking is fundamentally stronger than static block sparsity for approximating the underlying attention pattern, and how it avoids mixing unrelated tokens within blocks.
>
> * **Hyperparameters and generality.** We clarified that the two main knobs, maximum number of chunks $N_c$ and per-query budget $N_b$, are global sparsity budgets, not task-specific tuning. We use a single density (12.5%) across LongBench tasks and only vary it in Table 3 to study scaling behavior.
>
> * **Long-context and on-device deployment.** We already evaluate up to 100K tokens for 4-bit Llama-3.1-8B and report kernel-level latency up to 128K. We also added concrete CPU benchmarks (Intel Core 5 120U, PyTorch CPU) showing consistent latency gains over dense SDPA, to better support the on-device claim.
>
> * **Retention of important information.** We emphasized that DHSA’s dynamic segmentation groups semantically coherent spans, so chunk representations are dominated by meaningful tokens rather than positional blocks. Needle-in-a-Haystack at 100K context with only 6.25% preserved keys still achieves near-dense retrieval accuracy, indicating that key tokens are reliably retained even at extreme sparsity.
>
> * **“No retraining” clarification.** We clarified that DHSA never updates the base LLM: Transformer weights and architecture remain fixed. The only learned part is a small offline-trained boundary predictor shared across layers and tasks. Thus the base model is training-free, and DHSA can be used as a plug-and-play sparsity module.
>
> We hope this summary makes our contributions and clarifications more transparent and alleviates your remaining concerns.

---

### Official Review · Reviewer_SgM4 · 2025-10-31

**Soundness:** 2
**Presentation:** 3
**Contribution:** 2
**Rating:** 4
**Confidence:** 4

**Summary:**

This paper addresses the significant computational and memory costs (quadratic complexity) of the attention mechanism in long-context Large Language Models (LLMs), which is a major bottleneck, especially in resource-constrained environments.

Key Contributions：
* Dynamic Segmentation: DHSA first segments the input sequence into variable-length "chunks" based on the content itself. This is more adaptive than using fixed-size blocks.
* Hierarchical Computation: It then computes representations for these chunks using a special "length-normalized" method to avoid bias from different chunk sizes. It calculates similarity scores at this coarse, chunk-to-chunk level.
* Token-Level Upsampling: Finally, it upsamples these chunk-level scores to the token level to create an importance map. This map determines which fine-grained token-to-token attention scores are actually computed, preserving only the most impactful ones.

**Strengths:**

* Efficient long-context handling: Matches dense attention accuracy while cutting prefill latency by 20–60% and peak memory usage by 35% at 8K context, and scales to 100K context on a single 24 GB GPU (where dense kernels fail).
* Input-adaptive sparsity: Avoids rigid static patterns or heuristics; dynamically predicts attention sparsity via data-driven chunking and similarity, adapting to diverse tasks/inputs.
* Easy integration: Functions as a drop-in module for standard decoder-only Transformers, requiring no retraining or architecture changes to the base LLM.
* Robust chunk representation: Uses length-normalized aggregation to eliminate bias from variable chunk lengths, ensuring reliable similarity estimation.

**Weaknesses:**

* Hyperparameter dependence: Its performance relies on hyperparameters like the number of chunks and preserved keys, whose optimal settings vary across models, tasks, and hardware, lacking adaptive allocation strategies.

* Boundary predictor constraints: The boundary detector requires training on specific datasets (e.g., Long Data Collections) and may need adjustments for diverse text types, introducing potential generalization gaps.

* Hardware adaptability limitations: While tested on NVIDIA GPUs, its performance on other hardware (e.g., CPUs, edge devices) is not evaluated, raising questions about cross-hardware applicability.

**Questions:**

* In the ablation study, DHSA without dynamic chunking degrades to standard block-sparse attention, showing the critical role of dynamic chunking. However, the paper does not compare DHSA with recent advanced dynamic chunking methods (e.g., context-aware adaptive chunking). How does DHSA’s chunking strategy perform relative to these methods in terms of segmentation accuracy and computational efficiency?

* The boundary predictor uses soft labels derived from attention scores and focal BCE loss for training. If the base LLM itself has biased attention distributions (e.g., over-attending to trivial tokens), will this bias be transferred to the boundary predictor, affecting chunk quality? How to mitigate such potential bias propagation?

---

> ### Author Response · Authors · 2025-11-26
> **Response to Reviewer SgM4 (Part 1)**
>
> We thank the reviewer for the careful reading and constructive feedback. We clarify that DHSA’s two main hyperparameters, the maximum number of chunks $N_c$ and per-query token budget $N_b$, simply control the accuracy–efficiency trade-off, are not tuned per task, and yield robust performance across a wide range of sparsity ratios. We further strengthen the discussion and experiments on generalization: a single, compact boundary predictor trained on several QA and summarization corpora is reused across all benchmarks (including Needle-in-a-Haystack and LongBench) without per-dataset adaptation, and we additionally evaluate DHSA on CPU, showing consistent prefill speedups over dense attention. We clarify that the boundary predictor can be supervised using soft labels derived from a stronger teacher model’s attention, so biases in the base model’s attention do not directly propagate to the predictor. Finally, we position DHSA relative to context-aware adaptive chunking methods such as DCS: these approaches rely on external encoders and task-specific classifiers; they are not directly comparable at the attention level.
>
> ### **Response to Weakness 1**
>
> The two hyperparameters (the maximum number of chunks $N_c$ and the per-query token budget $N_b$) play the same role as the sparsity budget. They control the accuracy–efficiency trade-off rather than requiring task-specific fine-tuning of the model. In our experiments, we do not tune these hyperparameters per task. For LongBench, we use a single sparsity ratio (12.5%) across all tasks and models, and only vary it in Table 2 to study behavior at 6.25%, 12.5%, and 25%. As Table 2 shows, DHSA’s performance changes smoothly and monotonically with the sparsity ratio and remains better than baselines, indicating that it is robust to a wide range of settings and does not require extensive hyperparameter search.
>
> In deployment, $N_c$ and $N_b$ can be chosen directly from resource constraints (e.g., target latency or available memory): tighter budgets correspond to higher sparsity, while larger budgets allow DHSA to approach dense attention. Designing fully automatic budget selection policies (e.g., auto-tuning based on hardware/runtime feedback) is an interesting direction, and we clarify this as future work.
>
> ### **Response to Weakness 2**
>
> We explicitly evaluate cross-domain generalization of the boundary predictor. In all experiments, we train a single predictor on sequences from Long Data Collections, TriviaQA, and ChatQA2 (primarily QA and summarization), and then reuse it as is for all evaluation benchmarks, including Needle-in-a-Haystack and LongBench, which spans 16 heterogeneous datasets (single- and multi-doc QA, summarization, synthetic tasks, and code). We do not perform any per-dataset fine-tuning of the predictor, yet DHSA consistently matches dense attention across these diverse settings, suggesting that the local attention-pattern features used for boundary detection transfer well across text types.
>
> Moreover, the labels are generated automatically and the predictor is very small, so if a deployment scenario involves highly specialized domains, re-training or lightly adapting the predictor is inexpensive compared to tuning the LLM itself.
>
> ### **Response to Weakness 3**
>
> To demonstrate that DHSA generalizes beyond GPUs, we additionally benchmark kernel-level prefill speedup over dense attention (Torch.SDPA) on an Intel Core 5 120U CPU (Figure 8). For CPU experiments, we use an Intel Core 5 120U processor (10 cores, 12 threads, max frequency 1.40 GHz) running Windows 11, with Python 3.11, PyTorch 2.9.1+cpu, and Transformers 4.52.3. Across all evaluated sequence lengths and token densities, DHSA consistently reduces prefill latency relative to dense Torch.SDPA, showing that our sparsity predictor transfers beyond GPU kernels and remains effective on standard CPU hardware.
>
> ### **Response to Question 1**
>
> We appreciate the suggestion to compare with recent context-aware adaptive chunking approaches such as dynamic chunking and selection (DCS) [1]. These methods are closely related in spirit but operate at a different level of the pipeline, making a direct comparison less fair and substantially more involved. In particular, DCS-style methods use an additional encoder (e.g., Sentence-BERT) and a question-aware classifier to preprocess the input and select a subset of chunks. By contrast, DHSA is designed as a drop-in sparse attention module inside the LLM that does not rely on external encoders or task-specific classifiers, and can be applied uniformly to all LongBench tasks. We therefore focus on methods that are directly comparable at the attention level, which we believe is more appropriate for the scope of this paper.
>
>
> ## Reference
>
> [1] Sheng, Boheng, et al. "Dynamic Chunking and Selection for Reading Comprehension of Ultra-Long Context in Large Language Models." arXiv preprint arXiv:2506.00773 (2025).

---

> > ### Author Response · Authors · 2025-11-26
> > **Response to Reviewer SgM4 (Part 2)**
> >
> > ### **Response to Question 2:**
> >
> > The goal of DHSA, as with other sparse attention methods, is to preserve as much of the original attention mass as possible. In our setting, we treat the dense attention of the base model as the target. If the base LLM itself has biased attention distributions, we mitigate the issue by deriving soft boundary labels from the attention patterns of a stronger teacher model and reuse these labels across different base models. Thus, any bias in the base model’s attention does not influence the boundary supervision and cannot be directly propagated to the predictor.

---

> > > ### Comment · Reviewer_SgM4 · 2025-11-27
> > > **Official Comment by Reviewer SgM4**
> > >
> > > Thank you for the rebuttal. The authors have successfully addressed my primary concern. Accordingly, I have raised my score.

---

> > > > ### Author Response · Authors · 2025-11-27
> > > > **Response to Reviewer SgM4**
> > > >
> > > > Thank you for taking the time to review our response and reassess our submission! We are glad that our rebuttal addressed your concerns, and we remain committed to further improving the work!

---

### Official Review · Reviewer_fZ8k · 2025-11-01

**Soundness:** 3
**Presentation:** 4
**Contribution:** 2
**Rating:** 2
**Confidence:** 4

**Summary:**

The paper proposes Dynamic Hierarchical Sparse Attention (DHSA), an inference-time, drop-in sparse attention module for decoder-only LLMs. DHSA first dynamically chunks the sequence via a learned boundary predictor, then builds length-normalized chunk representations, computes chunk-chunk similarities, upsamples these scores back to the token level, and finally selects Top-Nb token interactions for each query. The method targets on-device settings and reports LongBench accuracy competitive with dense attention but having lower latency.

**Strengths:**

1. This paper tries to tackle an important problem of how to improve the efficiency of LLM inference in long context by leveraging sparsity in attention.

2. A clear hierarchical routing formulation with a concrete sparse attention pipeline. The design and implementation details are explained well.

3. The paper reports accuracy improvements over existing static sparse attention baselines and lower latency over full dense attention.

**Weaknesses:**

1. Missing comparisons to other more recent dynamic sparse baselines. Current baselines are mostly static patterns on static template.

2. Missing upper bound analysis with oracle top-k baseline to show how close the number of tokens selected is to the optimal choice. Missing latency comparison with baselines other than dense attention.

3. Still not clear why dynamic chunking is needed if there is an accurate way to estimate the contribution of each chunk to the overall attention.

4. Not clear how the system performs under batching settings.

**Questions:**

The paper did a comprehensive analysis and evaluation with static sparse attention baselines, including StreamingLLM, MInference and Block Sparse, but misses important dynamic sparse attention baselines. For example, [MagicPig](https://arxiv.org/abs/2410.16179) uses LSH sampling to select tokens for attention computation dynamically. [Quest](https://arxiv.org/abs/2406.10774) exploits query-aware sparsity that keeps track of minimal and maximal key values in the KV cache and estimates importance based on queries. Without a comparison with these state-of-the-art sparse attention baselines, it is hard to fully evaluate the benefits of the proposed approach.

It is not clear why dynamic chunking is needed, even though an ablation is provided. The ablation shows cases of DHSA without robust chunk representation and without dynamic and robust chunk representation. However, to demonstrate that dynamic chunking is indeed needed, it should further evaluate the case of DHSA with robust chunk representation and without dynamic chunking. Robust chunk representation is a normalized prefix sum for queries and keys in the chunk and should work independently of the chunk size selected. Also, I can imagine there are other ways to estimate the chunk similarity, for example, based on different clustering methods. However, the paper does not provide an evaluation of them other than the normalized prefix sum one.

There are also some evaluations missing in the paper. For example, it should provide an upper-bound analysis compared with the oracle top-k baseline on the number of tokens selected. In addition, performance numbers for the batching scenario are not evaluated.

1. How does DHSA perform in terms of accuracy compared to dynamic sparse attention baselines under the same sparsity setup?

2. Can you give some intuitions on why the boundary predictor is designed as this? For example, why is the left and right window not overlapped?

3. Can you show a comparison of DHSA with robust chunk representation and without dynamic chunking as an ablation? Have you evaluated other methods that could estimate the chunk similarity other than the current approach?

4. Can you provide the evaluation performance of DHSA under the batching scenario?

5. How are the ratios and hyperparameters of the baselines selected? Can you provide latency comparison with baselines? Can you provide number of tokens selected in the optimal case?

6. In Table 2 under sparsity = 25\%, how can DHSA outperform dense attention on LongBench Synth by such a large margin?

7. In Table 3 why is more memory needed for DHSA? Is it for storing the model weights of the boundary predictor?

---

> ### Author Response · Authors · 2025-11-25
> **Response to Reviewer fZ8k (Part 1)**
>
> We thank the reviewer for the constructive feedback and have substantially revised the manuscript to address these concerns. We now include additional dynamic baselines (DuoAttention, SeerAttention, and Quest), and report both accuracy (Tables 2, 3) and kernel-level prefill latency comparisons (Figure 8). To better characterize how close DHSA comes to the Top-K oracle, we strengthen our upper-bound analysis by (i) adding a Top-K overlap figure (Figure 12), and (ii) providing a theoretical analysis that formalizes DHSA’s advantage over block sparse attention (Appendix C). We clarify the role of dynamic chunking, add an ablation that isolates its effect (Table 4), and discuss why more complex clustering-based alternatives are less practical for sparse attention. Finally, we add batch-inference experiments (Figure 10), detail how baseline hyperparameters and densities are selected (Appendix D), and clarify both the Synth results (Table 3) and the modest memory overhead from DHSA’s routing buffers (Table 5). Together, these revisions directly address the raised weaknesses while reinforcing DHSA’s strengths in accuracy, efficiency, and robustness.
>
> ### **Response to Weakness 1 (dynamic baselines)**
>
> We have added comparisons to DuoAttention, SeerAttention, and Quest (Tables 2, 3). We note that DuoAttention and Quest primarily target KV-cache compression during decoding, which is orthogonal to DHSA’s focus on reducing prefill FLOPs.
>
> Empirically, we observe that DHSA performs particularly well in high-sparsity regimes. Compared with these baselines, DHSA wastes fewer keys when the attention budget is tight. Our dynamic chunking first aligns chunks with semantic segments, and then applies token-level Top-K selection within those segments. As a result, each query’s limited budget is spent on genuinely important tokens rather than on entire coarse blocks or fixed template positions. In contrast, template-based patterns must retain many uninformative tokens whenever a structural position is selected. Consequently, under high sparsity, their effective recall of important keys drops more than DHSA’s, consistent with our empirical results (Figure 3(b), Figure 12) and theoretical analysis (Appendix C).
>
> Specifically, DuoAttention performs head-wise selection between retrieval and streaming heads; once a head is classified as streaming, its sparsity is essentially static within that head. It does not learn a query-level dynamic sparsity mask. SeerAttention predicts block-wise sparsity gates on top of block-sparse attention, but still operates at the block level. Quest does perform dynamic sparsity, but at the page level and is designed primarily for KV-cache compression during decoding, rather than for reducing prefill FLOPs. This design difference explains why DHSA provides much stronger prefill acceleration while exceeding their accuracy at matched sparsity. In particular, we found DuoAttention is highly sensitive to the sparsity level: it performs well near its predefined sparsity of 0.5 but degrades sharply as the sparsity increases.
>
> ### **Response to Weakness 2 (upper bound analysis and latency vs baselines)**
>
> We have strengthened the analysis of how closely DHSA approaches the oracle Top-K selection. First, Figure 3(b) already shows that DHSA achieves higher attention recall than other sparse patterns and closely tracks the Top-K curve. To make this connection explicit, we additionally introduce Overlap with Top-K Attention comparison (Figure 12).
>
> To quantify how well different patterns preserve important tokens,  for each method, we compute the overlap between its retained keys and the Top-K targets. We report the average overlap to obtain a stable estimate. As shown in Figure 12, DHSA consistently achieves the highest overlap with the oracle Top-K set across all retention budgets, while other sparse patterns recover substantially fewer important tokens. Overall, DHSA allocates its sparsity budget more effectively and provides the closest approximation to dense attention.
>
> In addition, we add a theoretical analysis (Appendix C) that explains why DHSA outperforms block-sparse attention. We show that (i) by adapting chunk boundaries to semantic structure and using length-normalized chunk scores, DHSA can recover a near-oracle fraction of the important tokens, whereas (ii) fixed block grids are oblivious to semantics and must include many uninformative tokens whenever segments are misaligned with block boundaries. This formalizes DHSA’s advantage and aligns with the empirical Top-K overlap results.

---

> ### Author Response · Authors · 2025-11-25
> **Response to Reviewer fZ8k (Part 2)**
>
> We now provide kernel-level prefill latency comparisons against other sparse baselines in Figure 8. The figure reports speedup for StreamingLLM, MInference, Block-Sparse Attention, DuoAttention, SeerAttention, Quest, and DHSA. Among these methods, MInference’s vertical-slash pattern remains slower than FlashAttention-2, whereas DHSA yields substantial acceleration and approaches the speedup achieved by Block-Sparse Attention. In contrast, StreamingLLM, DuoAttention, and Quest are mainly designed for KV-cache compression and therefore provide no prefill speedup.
>
> ### **Response to Weakness 3 (necessity of dynamic chunking and alternatives)**
>
> The effectiveness of chunk similarity estimation is tightly coupled to the underlying segmentation. With fixed-size blocks, a single block often covers multiple sentences or topics, so its representation mixes unrelated tokens and produces noisy similarities. Our dynamic chunking explicitly detects semantic shifts, so each chunk is closer to a coherent span (e.g., a sentence, paragraph, or code block). This leads to much cleaner chunk-level similarities.
> We add an ablation that isolates the effect of dynamic chunking (Table 4) and theoretical analysis (Appendix C).
>
> Alternative designs based on clustering are computationally unattractive. Computing global token–token similarities to derive chunk assignments has essentially the same $O(L^2)$ cost as dense attention, eliminating any efficiency gains. Local similarity-based clustering (e.g., hierarchical clustering or K-means) still requires repeated similarity computations and iterative updates, leading to substantial overhead in practice for long sequences.
>
> Instead, our design can be viewed as a lightweight, supervised 1D segmentation mechanism tailored to attention. The boundary predictor is trained on attention-derived labels, so chunks are placed at positions where the model’s own attention patterns change. This produces contiguous, semantically coherent spans. Our experiments show that the proposed dynamic chunking provides substantial gains over static blocks for a given chunk similarity estimator (Table 4).
>
> ### **Response to Weakness 4 (batching behavior)**
>
> We have added batch-inference experiments (Figure 10). Batch inference is inherently challenging for dynamic sparse attention, since enforcing a shared sparsity mask across all sequences in a batch is often suboptimal. Instead of forcing such a shared mask, DHSA processes each sequence sequentially within a lightweight for-loop. Empirically, this per-example batching strategy is faster than batched FlashAttention-2 (FA2) and avoids the Out-of-Memory (OOM) failures that batched FA2 frequently encounters at long context lengths.
>
>
> ### **Response to Question 1 (dynamic baselines)**
>
> Our evaluation is not limited to static attention baselines. Block-Sparse Attention and MInference both predict sample-specific sparsity using fixed templates (e.g., block-wise or vertical-slash patterns). MagicPig and Quest, in contrast, primarily target KV-cache compression rather than KV generation, making them orthogonal, and in principle complementary, to DHSA. In the revision, we additionally report results for DuoAttention, SeerAttention, and Quest (Tables 2, 3). DHSA consistently achieves higher accuracy in high-sparsity regimes.
>
> ### **Response to Question 2 (boundary prediction design)**
>
> Our goal is to detect whether there is a boundary (chunk end) at tokens $i$. For this reason, we intentionally use non-overlapping left and right windows, namely $[k_{i-w+1}, \ldots, k_i]$ and $[k_{i+1}, \ldots, k_{i+w}]$. This design compares the content on the two sides of the candidate cut without mixing them. If the two windows are semantically similar, we do not place a boundary; if they differ sharply, we mark a boundary.
>
> Allowing the windows to overlap would cause many tokens to appear in both windows, which blurs this contrast: even at a true boundary, the two windows would share a large portion of their content, making their representations more correlated and the change harder to detect. We found that non-overlapping windows provide a cleaner "left vs. right" signal, and they align naturally with our automatic labeling scheme, which also uses non-overlapping windows.
>
> During development, we also explored with alternative architectures, including CNN-based variants (that take both sides as input), but ultimately found that the current configuration consistently achieved higher accuracy.

---

> ### Author Response · Authors · 2025-11-25
> **Response to Reviewer fZ8k (Part 3)**
>
> ### **Response to Question 3 (chunk similarity)**
>
> We have added an ablation that evaluates DHSA with robust chunk representation but without dynamic chunking (Table 4).
>
> When we disable dynamic chunking and use uniform blocks of size $B$, our chunk representations reduce to scaled averages over the tokens in each block, and the resulting chunk-level similarity are proportional to the block-wise similarity used in standard block-sparse attention. As a result, the variant without dynamic chunking effectively behaves like a conventional block-sparse pattern (see Appendix E for equations).
>
> For chunk similarity estimation, we explored (i) average pooling, (ii) max pooling, and (iii) our length-normalized average pooling. Length-normalized average pooling consistently achieved the best empirical performance, especially when dynamic chunking induces variable-length segments.
>
> As for clustering-based methods, token-level global similarity is as expensive as dense attention, while our method can be viewed as a lightweight, supervised 1D segmentation mechanism tailored to attention patterns, which preserves contiguity and remains efficient for long contexts.
>
> ### **Response to Question 4 (batching)**
>
> We have added batch-inference experiments (Figure 10). As noted above, batch processing is challenging for dynamic sparse attention, since forcing a common sparsity mask across all sequences in a batch is often suboptimal. Instead, DHSA processes each sequence in a lightweight for-loop, which empirically is faster than batched FlashAttention-2 and avoids the OOM issues that batched FA2 encounters at long sequence lengths.
>
> ### **Response to Question 5 (baseline hyperparameters and latency)**
>
> For a fair comparison, we use the same token density $N_b / L$ for all methods (6.25%, 12.5%, 25%), as reported in Table 3. For each baseline, we follow hyperparameter choices recommended in the original papers or official implementations, adjusting them only to match the target density.
>
> Specifically, we configure the baselines as follows:
>
> 1. **StreamingLLM**, corresponding to the *A-shape* pattern.
>    We allocate 20% of the preserved keys as global tokens and 80% as local windows.
>
> 2. **StreamingLLM with dilated windows**, which applies dilated local windows with fixed intervals.
>    We use 20% global tokens and 80% dilated windows with interval 1.
>
> 3. **StreamingLLM with strided windows**, which combines local windows with dilated attention.
>    We use 20% global tokens, 40% local windows, and 40% dilated windows with interval 1.
>
> 4. **MInference**, which supports *A-shape*, *Vertical-Slash*, and *Block-Sparse* patterns.
>    For clarity, we adopt the Vertical-Slash pattern, using 50% vertical-line tokens and 50% slash-line tokens, with last\_q = 64.
>
> 5. **Block-Sparse Attention**, which selects tokens via block-wise similarity but lacks dynamic chunking and robust block representations.
>    We use a block size of 128 in all experiments.
>
> 6. **DuoAttention**, which allocates retrieval heads and streaming heads to build a sparse KV cache.
>    The head mask is learned during training, and we adjust the sparsity ratio to match the desired density.
>
> 7. **SeerAttention**, which learns attention gates on top of block-sparse attention, with the key component being prediction of the block mask.
>    Because it does not natively support a fixed per-query token budget for sparse prefill, we set the non-zero ratio to match the target average density.
>
> 8. **Quest**, which compresses the KV cache to reduce memory footprint and improve decode-time efficiency.
>    Given a token budget, it restricts each query to attend to that number of tokens during decoding. We set this budget to match the effective density used in our setting.
>
> For StreamingLLM and its variants, MInference, Block-Sparse Attention, and Quest, we configure the patterns so that each query selects approximately the same number of tokens implied by the token density (up to small discrepancies due to block- or window-based selection).
> For SeerAttention, the non-zero ratio is chosen to match the same global density, though the exact number of tokens per query can vary.
> Note that DuoAttention effectively uses more tokens than other methods, since its local heads still attend to attention sinks and a fixed set of recent tokens.

---

> ### Author Response · Authors · 2025-11-25
> **Response to Reviewer fZ8k (Part 4)**
>
> We also provide kernel-level prefill latency comparisons against these sparse baselines in Figure 8. The figure reports speedup over FlashAttention-2 for StreamingLLM, MInference, Block Sparse, DuoAttention, SeerAttention, Quest, and DHSA. Among these methods, MInference’s vertical-slash pattern remains slower than FlashAttention-2, whereas DHSA yields substantial acceleration and approaches the speedup achieved by Block-Sparse Attention. In contrast, StreamingLLM, DuoAttention, and Quest are primarily designed for KV-cache compression and thus offer no prefill speedup.
>
> ### **Response to Question 6 (performance)**:
>
> The “Synth” column in Table 3 averages two datasets: PassageCount and PassageRetrieval_en.
> For Llama-3.1-8B, dense attention obtains 15.0 / 72.9 on these two datasets (avg 44.0), while DHSA at 25% density obtains 11.5 / 93.9 (avg 52.7). Thus the improvement mainly comes from PassageRetrieval_en.
>
> PassageRetrieval_en is essentially a within-context retrieval task: the model sees 30 candidate paragraphs plus an abstract and must output "Paragraph k". In this setting, dense attention is not an upper bound on accuracy: allowing the query to attend to all 30 paragraphs can introduce significant distraction from similar but incorrect paragraphs.
> DHSA’s content-adaptive sparsity acts as a soft retrieval / denoising step: it first groups tokens into paragraph-aligned chunks, then focuses attention on only the top-scoring chunks and their tokens.
> Empirically, this concentrates the attention budget on a few highly relevant paragraphs and suppresses many distractors, which improves the final classification accuracy.
>
> By contrast, in PassageCount the model must aggregate information across the whole context, where pruning can remove useful occurrences; here DHSA slightly underperforms dense, which is consistent with the intuition above.
>
> This phenomenon (sparse method performs better than dense attention on certain tasks) is also observed in literatures (e.g., MInference and SeerAttention).
>
> ### **Response to Question 7 (memory usage)**:
>
> Table 5 reports peak memory usage during prefill. DHSA shows slightly higher memory than dense attention because our kernel allocates several temporary buffers for sparsity prediction: (i) chunk-level representations, (ii) chunk-level similarity scores, and (iii) token-level importance scores and the resulting sparsity mask.
> These are ephemeral working tensors used only during the routing step.
> The additional parameters of the boundary predictor themselves are tiny compared to the base LLM and have negligible impact on the overall memory footprint.
>
> This kind of internal overhead is not unique to DHSA, other dynamic sparse methods (e.g., MInference, SeerAttention) also maintain auxiliary scores, masks, or gating networks on top of the KV cache.
>
> In our setting, the extra memory is modest and does not change the asymptotic scaling with context length: for long sequences, the KV cache and MLP activations remain the dominant contributors to memory consumption.
> In return for this small temporary overhead, DHSA substantially reduces attention FLOPs and latency, yielding a favorable accuracy–efficiency trade-off.

---

> ### Author Response · Authors · 2025-11-28
> **Response to Reviewer fZ8k (Part 5)**
>
> ### **Additional Response to Question 3 (clustering-based methods).**
>
> We provide a more detailed comparison between our dynamic chunking and clustering-based approaches. An alternative is to use K-means to cluster keys into clusters [1]. **Tactic** [1] focuses on **decoding**: during prefill it runs K-means on keys per head/layer to obtain cluster centroids and assignments, and at decode time, for a query $q$, it (i) computes $q \cdot c_j$ for each cluster centroid $c_j$, (ii) sorts clusters by this score ("cluster criticality"), and (iii) unpacks tokens cluster by cluster to obtain a partially sorted token list.
>
> Suppose we adapt this idea to sparsify **prefill**: for each query in prefill, we would (i) compute scores to the cluster centroids, (ii) pick the topK clusters for that query, and (iii) run dense attention only within those clusters, i.e., only over tokens whose keys lie in the selected clusters.
>
> We argue that this approach is less preferable than DHSA for **prefill** speedup:
>
> 1. **Computation and GPU-friendliness.**
>    Using K-means as the basis for prefill sparsification requires running multiple iterations of K-means over all $L$ keys per head/layer. Each iteration has cost $O(L \cdot C \cdot d_k)$, where $L$ denotes sequence length, $C$ is the number of clusters and $d_k$ is the key dimension. With a roughly fixed average cluster size (e.g., 32~64 tokens per cluster, so $C \propto L$), this scales **quadratically in the sequence length** $O(L^2)$. In contrast, DHSA uses a single lightweight boundary predictor (MHA + feature fusion + MLP, ~20 MB, **local context only** $O(L)$) that is implemented with standard attention and linear layers, making it more GPU-friendly and faster than iterative K-means as the sequence length increases.
>
> 2. **Length-normalized chunk representations vs plain centroids.**
>    K-means represents each cluster by the **unscaled centroid** (plain arithmetic mean of keys), whose norm and semantic sharpness degrade as the cluster becomes larger and more heterogeneous. Our chunk representation instead uses **length-normalized** average pooling: chunk similarities are approximately invariant to the chunk length, so short and long segments compete on a comparable scale. This is important for chunk-level routing, since long but relevant segments are not systematically downweighted. In contrast, centroid-based scores are sensitive to cluster cardinality and can become blurred when many diverse tokens are grouped together.
>
> 3. **Non-contiguous clusters and causal masking complexity.**
>    K-means produces clusters that are non-contiguous in the sequence dimension: tokens from a single cluster can be scattered across the entire context. To use such clusters for prefill attention, we must either **process each query independently** (as in Tactic’s decoding) or **gather queries and keys per cluster**. Both options lead to **irregular** memory access patterns and complicate causal mask construction, making the resulting kernel significantly more complex than our contiguous chunk-based design.
>
> 4. **Compatibility with Flash attention-2.**
>    DHSA’s dynamic chunking yields contiguous segments, and the resulting sparsity pattern is directly **compatible** with efficient **Flash attention-2 (block-sparse) kernels**. In our experiments, block-sparse kernel is substantially faster than other baselines (Figure 8). In contrast, non-contiguous clusters cannot be mapped cleanly to blocks, and therefore cannot fully exploit block-sparse implementations during prefill.
>
> Taken together, these considerations suggest that while K-means–based clustering is useful for **structuring the KV cache** (as in [1]), using it as the primary mechanism for **prefill** sparsification is less practical and less efficient than our design.
>
>
> ## Reference
>
> [1] Zhu, Kan, et al. "Tactic: Adaptive sparse attention with clustering and distribution fitting for long-context llms." arXiv preprint arXiv:2502.12216 (2025).

---

> ### Author Response · Authors · 2025-12-03
> **Additional clarification for Reviewer fZ8k**
>
> As the discussion period concludes, we would like to briefly summarize how we addressed your main concerns:
>
> * **Dynamic sparse baselines.**
>   We added **DuoAttention**, **SeerAttention**, and **Quest** as dynamic/learned sparsity baselines (Tables 2 and 3), and reported **kernel-level prefill latency** for all sparse methods including StreamingLLM, MInference, Block-Sparse, DuoAttention, SeerAttention, Quest, and DHSA (Figure 8). Under matched sparsity, DHSA consistently achieves higher accuracy in high-sparsity regimes and provides substantial **prefill speedup over FlashAttention-2**, approaching the speed of Block-Sparse Attention, while several dynamic baselines (DuoAttention, Quest) mainly target KV-cache compression and provide no prefill gains.
>
> * **Upper bound and oracle comparison.**
>   Beyond the original recall curve (Figure 3(b)), we added an **overlap-with-Top-K** analysis (Figure 12). DHSA has the highest overlap with the Top-K set across retention budgets, indicating its selections are closest to oracle. We also provide a theoretical analysis (Appendix C) explaining why semantically aligned, length-normalized chunks yield a better approximation than fixed block grids, which must retain many uninformative tokens when boundaries are misaligned.
>
> * **Motivation for dynamic chunking (vs. static blocks / clustering).**
>   We added an ablation that isolates **robust chunk representation without dynamic chunking** (Table 4): with uniform blocks, DHSA effectively collapses to block-sparse behavior and loses performance. Dynamic chunking ensures chunks track semantic shifts, producing cleaner chunk similarities for the same estimator. We also discuss clustering-based alternatives (e.g., K-means approaches, Tactic) and argue they are less suitable for **prefill**: they either incur $O(L^2)$ costs, produce non-contiguous clusters that are unfriendly to causal masks and FlashAttention-style kernels, or blur semantics when cluster sizes grow.
>
> * **Batching behavior.**
>   We added **batch-inference experiments** (Figure 10). Rather than enforcing a shared sparsity mask across batch elements, DHSA processes each sequence in a lightweight for-loop. Empirically, this strategy is still faster than batched FlashAttention-2 at long context lengths and avoids OOM issues that FA2 encounters in these regimes.
>
> * **Hyperparameters, latency, and optimal token counts.**
>   All methods are compared under the same **global density** (6.25%, 12.5%, 25%), and we detail how each baseline’s windows/blocks/gates are set to match that density (Appendix D). Figure 8 reports **prefill latency speedups** for all baselines. The Top-K overlap analysis quantifies how many optimal tokens each method recovers as a function of budget.
>
> * **Clarifying Synth-task behavior and memory overhead.**
>   We clarified that the Synth task gain at 25% sparsity is driven by PassageRetrieval_en, where adaptive sparsity works as a denoising / soft retrieval mechanism and can outperform dense attention by suppressing distractor paragraphs. For memory (Table 5), DHSA’s modest overhead comes from temporary routing buffers; the boundary predictor itself is tiny. This does not change the asymptotic scaling, and in practice the KV cache and MLP activations still dominate memory, while DHSA provides clear latency benefits and further reduces MLP activation memory for very long sequences.
>
> We hope this summary clarifies the motivation and empirical support for dynamic chunking, the fairness and strength of our comparisons, and the practical behavior of DHSA in real deployment settings.

---

### Official Review · Reviewer_CFPb · 2025-11-01

**Soundness:** 3
**Presentation:** 3
**Contribution:** 3
**Rating:** 4
**Confidence:** 4

**Summary:**

Due to the quadratic nature of attention, prefill attention becomes the key bottleneck for long-context inference. Existing systems prune tokens based on heuristics or pre-chosen patterns, which limits accuracy. This paper proposes **DHSA**, which trains an MLP layer to dynamically predict the boundaries of token chunks and uses dot similarity to model interactions between chunks. The actual attention is then computed only on highly relevant chunk pairs. DHSA achieves good accuracy and speedups on long-context inference.

**Strengths:**

- The dynamic partitioning of tokens is a novel and effective idea.
- The accuracy evaluation results look promising.

**Weaknesses:**

- DHSA requires training an MLP layer to predict chunk boundaries, making it harder to deploy than existing methods.
- The efficiency evaluation is not very comprehensive.

**Questions:**

Thanks for submitting to ICLR 2026. The paper introduces an interesting idea of dynamically partitioning sequences to group similar tokens into the same chunk. However, I still have some concerns about the paper.

- Firstly, since DHSA involves training, it would be more fair to compare it with other methods that also train a small predictor, such as DSA or SeerAttention. These should provide much stronger performance than the current baselines. Additionally, DuoAttention may also be a good baseline, especially at 12.5% or 25% sparsity.

- Moreover, the upsampling process seems to violate the assumption of partitioning. Theoretically, similar tokens should already be grouped together, and we should expect clear boundaries between chunks.

- Additionally, during MLP training, it is unclear what \( f_{MHA} \) represents. Which **Q** vector is being used in this computation?

- Regarding efficiency, the evaluation is based on PyTorch implementations. FlashAttention might be a better baseline for fair comparison. It is also unclear how to efficiently implement block-sparse attention given the dynamic chunk sizes.

- For inference, since DHSA treats all newly generated tokens as a single chunk, what happens in long-generation tasks? Will this chunk grow too large and degrade performance?

---

> ### Author Response · Authors · 2025-11-25
> **Response to Reviewer CFPb (Part 1)**
>
> We thank the reviewer for the constructive feedback and have revised the paper accordingly. To strengthen the empirical comparison, we added DuoAttention and SeerAttention as learned sparsity baselines at matched sparsity levels (Tables 2, 3), and show that DHSA especially outperforms them in high-sparsity regimes, consistent with our analysis (Figure 3(b), Figure 12, Appendix C). We clarify that the boundary predictor is a compact module (~20 MB) trained once offline from automatically derived attention labels, shared across layers and datasets, and that DHSA can also operate with externally provided boundaries (e.g., line breaks, section markers), in which case the boundary predictor is not required. To address efficiency concerns, we expanded our evaluation to include kernel-level benchmarks against FlashAttention-2 on GPU and Torch.SDPA on CPU, and described two complementary kernels (PyTorch SDPA and FlashAttention-2) that make DHSA practical across models and hardware. Finally, we refined the technical exposition: we explain the upsampling process, clarify the definition and role of $f_{\text{MHA}}$, and detail the online boundary policy for long generation.
>
> ### **Response to Weakness 1 (training the boundary prediction MLP)**
>
> We agree that DHSA introduces a small learned component for boundary prediction, but we view this as a light, one-time cost rather than a deployment barrier:
>
> * **Small, shared predictor trained offline.**
>   The boundary predictor is a compact 2-layer MLP with MHA-based encoders of total size ~ 20 MB, trained once per base model using automatically derived labels from the model’s own attention patterns (no human annotation). The predictor is shared across all layers and all evaluated datasets, and training operates on a few public long-context corpora (Long Data Collections, TriviaQA, ChatQA2). After training, deployment only requires loading this small module alongside the frozen base LLM.
>
> * **Not tied to a learned predictor.**
>   Importantly, DHSA is agnostic to how boundaries are obtained. When reliable structural cues exist (e.g., line breaks ‘\n’ / ‘\n\n’, section markers, JSON / markdown delimiters, code indentation), the framework can directly take these as chunk boundaries without using the predictor. Our current implementation already supports mixing the learned boundary scores with explicit boundaries (e.g., ‘\n\n’). In this mode, DHSA still performs hierarchical sparsification; the learned predictor is simply a way to improve the alignment between chunks and semantic segments.
>
> * **Accuracy benefit relative to static sparsity and other learned methods.**
>   As shown in our ablations (Table 4), removing dynamic chunking reduces average performance on LongBench from 31.8 to 28.0. The small predictor buys accuracy at the same density budget. We also note that many recent dynamic sparsity methods, such as DuoAttention and SeerAttention, train additional modules (head masks, block gates) jointly with the base LLM. In contrast, DHSA trains a lightweight, task-agnostic predictor separately, which makes it much simpler and more cost-effective.
>
> Overall, while DHSA does introduce a small learned boundary predictor, this component is (i) compact and shared, (ii) trained offline using self-supervised labels, (iii) optional when explicit boundaries are available, and (iv) substantially less expensive than the training required by other learned sparsity baselines, while providing clear accuracy gains over heuristic sparse patterns.
>
> ### **Response to Weakness 2 (efficiency evaluation)**
>
> We thank the reviewer for highlighting the need for a more comprehensive efficiency study, and we have expanded our experiments accordingly.
> We first report DHSA prefill speedup over FlashAttention-2 at the kernel level (Figure 7); across all evaluated sequence lengths and density levels, DHSA consistently yields lower latency. We then compare kernel-level prefill speedup against additional sparse attention baselines in Figure 8, where DHSA provides substantial acceleration and approaches the speedup of Block-Sparse Attention. To demonstrate that DHSA is not limited to GPUs, we also benchmark kernel-level prefill speedup over Torch.SDPA on an Intel Core 5 120U CPU (Figure 9). Implementation details can be found in Appendix D. Across all these settings, DHSA consistently accelerates the prefill stage.
>
> ### **Response to Question 1**
>
> We have added comparisons to DuoAttention and SeerAttention; the results are reported in Tables 2 and 3. Empirically, DHSA performs particularly well in high-sparsity regimes. For these baselines, when sparsity is high, their effective recall of important keys drops more than DHSA’s, which is consistent with our empirical results in Figure 3(b), Figure 12, and the theoretical analysis in Appendix C. In particular, DuoAttention is highly sensitive to the sparsity level: it performs well near its predefined sparsity of 0.5 but degrades sharply as the sparsity increases.

---

> ### Author Response · Authors · 2025-11-25
> **Response to Reviewer CFPb (Part 2)**
>
> ### **Response to Question 2**
>
> For the upsampling step, we copy each entry of the chunk-level similarity $S_c$ into the corresponding submatrix of the token-level similarity $S_t$ (see Figure 5). This preserves the partitioning assumption: for any token $x$ in chunk $A$ and token $y$ in chunk $B$, their score in the upsampled matrix is exactly the score between chunks $A$ and $B$.
>
> The motivation for upsampling is that chunk lengths vary, so we cannot directly apply chunk-level Top-K selection. Pure chunk-level selection admits whole chunks and may exceed the token budget. For example, if a single chunk contains 512 tokens while the budget is only 256 tokens, selecting that chunk would immediately overflow the budget. By upsampling to the token level and performing token-level selection, DHSA enforces the compute cap while preserving efficiency.
>
> ### **Response to Question 3**
>
> We apologize for the confusion. In Eq. (1), $f_{MHA}$ denotes a lightweight self-attention module applied over local key windows $k$. It uses its own projections $(W_Q^{(b)}, W_K^{(b)}, W_V^{(b)})$ to form $(q^{(b)}, k^{(b)}, v^{(b)})$ from those keys, i.e., $q^{(b)}=W_Q^{(b)}k$, $k^{(b)}=W_K^{(b)}k$, and $v^{(b)}=W_V^{(b)}k$. We then average-pools the outputs $o^{(b)} = \text{MultiHead}(q^{(b)}, k^{(b)}, v^{(b)})$ to produce $k_{\text{left}}$ and $k_{\text{right}}$. We do not reuse the base model’s $q$. We chose keys as inputs because our boundary labels are derived from attention-mass patterns; keys provide the most direct signal for these patterns.
>
> ### **Response to Question 4**
>
> We report DHSA prefill speedup over FlashAttention-2 at the attention-kernel level (Figure 7), and across all evaluated sequence lengths and density levels, DHSA consistently yields lower latency.
>
> We provide two kernel implementations tailored to different deployment scenarios (Appendix D), which together make DHSA practical in a wide range of settings:
>
> **1) Built on PyTorch SDPA.**
> This kernel is implemented in PyTorch with Hugging Face Transformers, leveraging PyTorch’s scaled dot-product attention (SDPA) backend. At each Transformer layer, the boundary predictor uses key states with positional embeddings to predict boundary indices, which partition the sequence into variable-length chunks. For each chunk, we employ prefix-sum aggregation to efficiently compute the sum of token queries and keys, then obtain the chunk representation by dividing this sum by the chunk length and normalizing with the chunk size. We compute the chunk-level similarity matrix and then expand this similarity to the token level by mapping each entry to the corresponding submatrix according to the chunk boundaries.
>
> For each *query chunk*, we perform *token-level Top-K selection* to identify a compact set of $N_b$ key tokens that satisfy causal constraints. The kernel then *gathers* the corresponding key/value tokens into *dense tiles* and constructs an exact causal mask using their absolute positions.
> For example, given the $i$-th query chunk $q_{b_i:b_{i+1}}$, where $b_i$ and $b_{i+1}$ denote the chunk boundaries, and a selected key-index set $J_i =$ {$j_{i,1}, j_{i,2}, \ldots, j_{i,N_b}$}, the kernel gathers the keys {$k_j \mid j \in J_i$} and performs *dense* attention between the *query chunk* and these *selected keys*. Since this operation is restricted to the small set of selected keys, it is both efficient and fast.
>
> **2) Built on FlashAttention-2.**
> Our method can also use Block Sparse Attention by predicting a block-sparse mask with minimal modification. When selecting boundaries, we restrict candidate chunk boundaries to block boundaries, i.e., {$0, B, 2B, 3B, \dots$}, where $B$ is the block size. Note that the generated chunk lengths remain variable, with boundaries at positions such as {$0, B, 3B, 7B, \dots$}. When computing chunk-level similarity, we then upsample to the *block level* and perform Top-K selection with $K = \lfloor N_b / B \rfloor$. Once the block-sparse mask is obtained, we directly invoke Block Sparse Attention to compute attention efficiently.
>
> Overall, the PyTorch SDPA implementation offers flexible, token-level selection and broad compatibility (e.g., Gemma 2/3, CPU), while the FlashAttention-2 implementation provides higher efficiency. These two backends complement each other in practice.

---

> ### Author Response · Authors · 2025-11-26
> **Response to Reviewer CFPb (Part 3)**
>
> ### **Response to Question 5**
>
> For long-generation tasks, we equip DHSA with an online boundary policy that decides where to close chunks as new tokens are generated. Concretely, the boundary predictor operates with a small $w$-token look-ahead: once token $i + w$ has been produced, it examines the local key windows $[k_{i-w+1}, \ldots, k_i]$ and $[k_{i+1}, \ldots, k_{i+w}]$ (using the same architecture in Figure 6). If the predicted boundary probability at position $i$ exceeds the threshold (after non-maximum suppression), we finalize a chunk ending at $i$ and start a new one at $i+1$. This streaming-style policy delays boundary decisions by at most $w$ tokens and does not require access to future context beyond this small local window.
>
> Empirically, on our evaluated tasks, the generated responses are short (non-summarization typically <128 tokens; summarization typically <512). Consequently, in our main results we adopt a simplified variant (Algorithm 2) that treats the entire generated prefix as a single chunk, and we do not observe any degradation in accuracy under this setting.

---

> ### Author Response · Authors · 2025-12-02
> **Additional clarification for Reviewer CFPb**
>
> As the discussion period concludes, we would like to briefly summarize how we addressed your concerns:
>
> * **Fairness of baselines / learned sparsity:**
>   We added **DuoAttention** and **SeerAttention** as learned sparsity baselines at matched sparsity levels (6.25%, 12.5% and 25%) in Tables 2 and 3. DHSA consistently matches or outperforms them, especially in high-sparsity regimes, which aligns with our analysis of better recall of important keys under tight budgets.
>
> * **Cost and practicality of the boundary MLP:**
>   The boundary predictor is a compact ~20 MB module, trained **once per base model** from automatically derived attention labels, and **shared across layers and datasets**. It is not jointly trained with the LLM and does not require any modification of base model weights. Moreover, DHSA can also operate using *external structural boundaries* (e.g., line breaks, section markers, code indentation), in which case the predictor is optional. Thus, deployment only adds a small, fixed auxiliary module, which we see as significantly lighter than the training overhead in many other learned sparsity methods.
>
> * **Partitioning and upsampling:**
>   The upsampling step copies each chunk–chunk score into the corresponding token–token submatrix, so every token pair across chunks inherits the same score as its parent chunks. This preserves the partitioning assumption at the *score* level, while enabling **token-level Top-K** to respect a fixed token budget under variable-length chunks.
>
> * **Clarification of $f_{\text{MHA}}$:**
>   We clarified that $f_{\text{MHA}}$ is a lightweight self-attention over **keys** only, with its own projections $(W_Q^{(b)}, W_K^{(b)}, W_V^{(b)})$. We do **not** reuse the base model’s query vectors; instead, keys are used because our training labels come from attention-mass patterns, for which keys are the most direct signal.
>
> * **Efficiency and FlashAttention:**
>   We added kernel-level **prefill latency** comparisons against **FlashAttention-2** on GPU and Torch.SDPA on CPU (Figures 7–9), and described two implementations:
>   (i) a PyTorch SDPA backend with token-level Top-K + gather, and
>   (ii) a FlashAttention-2 backend via predicted block-sparse masks.
>   Both backends show consistent prefill speedups over their dense counterparts, demonstrating that DHSA is compatible with modern efficient kernels.
>
> * **Long-generation behavior:**
>   For long generation, we described an **online boundary policy** with a small look-ahead window that incrementally closes chunks, preventing a single decode chunk from growing unbounded. On our benchmarks, generations are relatively short, so treating all generated tokens as one chunk already works well, but the streaming policy is available for genuinely long-horizon generation.
>
> We hope this summary clarifies the fairness of our comparisons, the practicality of deploying DHSA, and the soundness of the technical design.

---

### Note · Program_Chairs · 2026-01-22
**Submission Desk Rejected by Program Chairs**

Rebuttal Pdf exceeds 10 page limit.